# CROSS-CANCER KNOWLEDGE TRANSFER IN WSI-BASED PROGNOSIS PREDICTION

## ABSTRACT

Whole-Slide Image (WSI) is an important tool for estimating cancer prognosis. Current studies generally follow a conventional cancer-specific paradigm in which each cancer corresponds to a single model. However, this paradigm naturally struggles to scale to rare tumors and cannot leverage knowledge from other cancers. While multi-task learning frameworks have been explored recently, they often place high demands on computational resources and require extensive training on ultra-large, multi-cancer WSI datasets. To this end, this paper shifts the paradigm to *knowledge transfer* and presents the first preliminary yet systematic study on cross-cancer prognosis knowledge transfer in WSIs, called CROPKT. It comprises three major parts. (1) We curate a large dataset (UNI2-h-DSS) with 26 cancers and use it to measure the transferability of WSI-based prognostic knowledge across different cancers (including rare tumors). (2) Beyond a simple evaluation merely for benchmarking, we design a range of experiments to gain deeper insights into the underlying mechanism behind transferability. (3) We further show the utility of cross-cancer knowledge transfer, by proposing a routing-based baseline approach (ROUPKT) that could often efficiently utilize the knowledge transferred from off-the-shelf models of other cancers. CROPKT could serve as an inception that lays the foundation for this nascent paradigm, *i.e.*, WSI-based prognosis prediction with cross-cancer knowledge transfer.

## 1 INTRODUCTION

Whole-Slide Image (WSI) refers to the H&E stained histopathology image with extremely high resolution, *e.g.*, $40,000 \times 40,000$ pixels. It contains abundant visual information reflecting tumor progression, such as cellular morphology and tissue infiltration, and serves as the gold standard for cancer diagnosis. Owing to these, WSIs have been widely used for cancer prognosis—estimating the future survival of cancer patients (Kather et al., 2019a; El Nahhas et al., 2024; Liu et al., 2025). Accurate prognostic prediction has been demonstrated to be instrumental in guiding personalized decision-making and enhancing patient survival (Skrede et al., 2020; Chen et al., 2025).

To train "clinical-grade" prognostic models based on WSIs, the vast majority of existing studies (Chen et al., 2021; Liu et al., 2024a; Xu et al., 2025; Zhou et al., 2025; Wu et al., 2025) follow a cancer-specific approach in model development: (*i*) curating WSIs and follow-up labels from the patients (usually $N \approx 1,000$) with a specific cancer disease; (*ii*) fitting a specialized model with the training samples split from the curated dataset; (*iii*) finally, evaluating this model's performance on held-out test samples. This approach has shown considerable success (Lu et al., 2023), becoming a *de facto* paradigm for developing WSI-based prognostic models.

Nevertheless, this paradigm faces inherent limitations that hinder its ability to meet key practical requirements, as follows. **(1) Scaling to rare tumor diseases**. Similar to common diseases like lung or breast cancer, rare tumors also demand accurate prognosis to guide therapeutic planning. However, the study cohorts of rare diseases are often difficult to collect. Most of them comprise only a small number of patients or exhibit a high proportion of censorship. As a result, cancer-specific approaches often yield unsatisfactory prognostic models (Zadeh & Schmid, 2020; Lu et al., 2023) for rare tumor diseases. **(2) Benefiting from the generalizable prognostic knowledge of other cancers**. In clinical practices, for a specific cancer, the survival analysis cohort usually contains around 1,000 patients (Lu et al., 2022; Liu et al., 2024a). This often leads to models with weak generaliz-

ability (Song et al., 2024a). To mitigate this, it is highly anticipated that the prognostic knowledge from other cancers could be leveraged to enhance generalizability. However, current frequently-adopted schemes are generally cancer-specific—naturally incapable of learning from other cancers in training. This inherently limits the potential for further performance gains in state-of-the-art deep networks (Ilse et al., 2018; Chen et al., 2021; Shao et al., 2021).

Recent studies (Wulczyn et al., 2020; Vale-Silva & Rohr, 2021; Yuan et al., 2025) have explored multi-task learning (MTL)-based solutions. They treat each cancer as a single task and use multi-cancer datasets to train a single model to benefit from other cancers. However, this introduces an ultra-large WSI dataset and extensive, large-scale training, which require computational resources orders of magnitude greater than single-task learning. Besides, these studies naively group all available cancer types at hand in training, without considering the *underlying negative effect* of some cancers on the cancer of interest in task grouping (Standley et al., 2020; Fifty et al., 2021), thereby resulting in sub-optimal prognostic models.

Given these limitations, this paper proposes a paradigm shift from cancer-specific or multi-cancer learning to **knowledge transferring**[1] (Zhuang et al., 2020). (1) Knowledge transfer avoids training models directly on a very limited number of rare tumor samples; instead, it could utilize the underlying generalizable knowledge of other cancers for prognosis estimation. (2) Knowledge transfer offers a potential alternative to large-scale training on multi-cancer datasets by leveraging multiple off-the-shelf, fitted cancer-specific models or slide-level foundation models (Shao et al., 2025; Ding et al., 2025). This approach may emerge as a considerably more cost-efficient and effective strategy for harnessing insights from diverse cancer types. Despite these, knowledge transfer remains under-studied in WSI-based prognosis prediction, with many primary questions unanswered, *e.g.*, can and why can WSI-based prognostic knowledge be transferred across different cancers? can cross-cancer knowledge be transferred to improve prognostic performance?

With these primary questions in mind, this paper presents a preliminary yet systematic study on cross-cancer prognosis knowledge transfer in WSIs, named CROPKT. It comprises three major parts. **(1) Transferability evaluation**: we derive a large WSI dataset with 26 cancers (including rare tumor diseases, called UNI2-h-DSS) and then utilize this dataset to train prognostic models and measure these models' performance in cross-cancer transferring. **(2) Transferability insights**: beyond a simple benchmark that merely evaluates transferability, we further design a range of experiments to gain deeper insights into it: (*i*) what knowledge a transferred model can offer and (*ii*) what factors affect cross-cancer knowledge transfer. **(3) Transferability utility**: to demonstrate the utility of knowledge transfer in improving prognostic performance, we propose a routing-based baseline approach (called ROUPKT) that can adaptively combines the beneficial knowledge transferred from multiple off-the-shelf prognostic experts. We hope CROPKT could lay the foundation for this new paradigm, *i.e.*, WSI-based prognosis prediction with cross-cancer knowledge transfer. The key contributions of this paper are summarized as follows:

① This paper shifts the paradigm from cancer-specific and multi-cancer training to knowledge transferring. To our best knowledge, it presents for the first time a systematic study (*i.e.*, CROPKT) on cross-cancer knowledge transfer in WSI-based prognosis prediction.

② We curate a large WSI dataset (UNI2-h-DSS) for this study. It covers 26 cancer types and contains 11,188 WSIs from 9,190 patients, with complete follow-up labels for survival analysis and cancer-specific sub-datasets and stable data splits for performance benchmarking.

③ We quantify the transferability of WSI-based prognostic knowledge to common cancers and rare tumors. Beyond a simple benchmark that merely measure transfer performances, we further design a range of experiments to seek deeper insights into the underlying mechanism of transferability.

④ We show the utility of cross-cancer knowledge transfer in WSI-based prognosis prediction, by proposing a routing-based baseline approach, ROUPKT. Notably, an average improvement of 3.1% across 13 cancer types is obtained by cross-cancer knowledge transfer via ROUPKT. Dataset, code, and models will be released to facilitate future research.

---

[1]In this study, knowledge transfer refers to a process where a model trained on one cancer type is transferred and leveraged for a task involving another cancer type.

## 2 RELATED WORK

**WSI-based Cancer Prognosis** The core task of WSI-based cancer prognosis is to estimate the future survival of cancer patients using histological WSIs, falling within the scope of survival analysis (SA) (Wang et al., 2019) in methodology. To model gigapixel WSIs for SA, each image is usually processed into tens of thousands of patches and ultimately is taken as a bag of multiple instances, where each instance is a feature vector extracted from a patch by a pretrained foundation model like UNI (Chen et al., 2024). To learn the presentation of bags, multi-instance learning (MIL) algorithms are often adopted for aggregating multiple instances (Liu et al., 2024b). Most existing studies focus on devising better MIL schemes or multi-modal strategies, *e.g.*, GNN (Chen et al., 2021; Wu et al., 2022; Liu et al., 2023), Transformer (Shao et al., 2023; Yuan et al., 2025), GAN (Liu et al., 2024a), VAE (Zhou et al., 2025), MoE (Wu et al., 2025), prompt learning (Liu et al., 2025; Xu et al., 2025), *etc*. They have demonstrated remarkable success within this domain. However, they are still limited to cancer-specific or multi-cancer training in model development.

**Knowledge Transfer** Inspired by educational psychology, knowledge transfer is studied to realize the generalization of experience, often applied in scenarios where only a few samples can be used for learning (Zhuang et al., 2020). Since knowledge transfer does not always have a positive impact on target tasks, numerous efforts have been made to maximize the performance of transferring knowledge to a target domain, *e.g.*, DAN (Long et al., 2015), CORAL (Sun et al., 2016), and UAN (You et al., 2019). Nevertheless, very few works are seen in the field of WSI-based cancer prognosis. Although a recent study (Shao et al., 2025) has explored knowledge transfer, it focuses more on the transferability of MIL models in WSI-based classification tasks. The transferability of WSI-based prognostic knowledge is still unclear, with many fundamental issues to be addressed. Therefore, this paper's purpose is to present a preliminary yet systematic study on this nascent topic. Concretely, this paper aims to investigate whether and why WSI-based prognostic knowledge can be transferred across different cancers, as well as whether cross-cancer knowledge can be transferred to improve prognostic performance, instead of being dedicated to devising a better transfer learning approach with state-of-the-art transfer performance.

## 3 PRELIMINARIES

This section introduces the preliminaries of the whole study, including dataset curation and essential settings (illustrated in Figure 1). We show their details below.

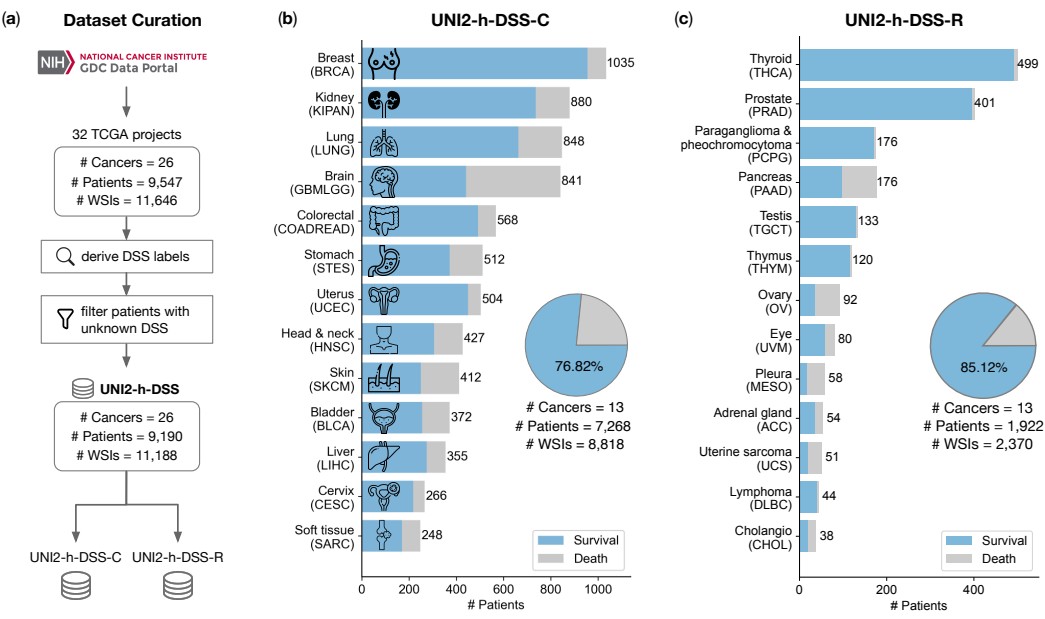

Figure 1: Dataset curation and statistics. C and R refer to common and rare cancer diseases.

**Dataset Curation** To study cross-cancer knowledge transfer in WSI-based prognosis, we derive a large WSI dataset with 26 cancer types, named UNI2-h-DSS. As shown in Figure 1(a), its curation consists of the following steps. **(1)** We first locate 9,547 patients with 11,646 diagnostic WSIs from 32 TCGA projects, following the setting of UNI (Chen et al., 2024). As some projects point to the same organ, we merge them into a larger one according to conventions. This leads to 26 cancer types. **(2)** Then, we derive DSS (disease-specific survival) labels from Liu et al. (2018) for all patients; DSS is chosen as the event of interest because its follow-up labels are of higher quality and it is more related to cancer-caused death (Liu et al., 2018). After filtering the patients with unknown DSS, there are 9,190 patients with 11,188 WSIs and 26 cancer types retained for this study. All WSIs are processed into patch features by UNI2-h (Chen et al., 2024), a state-of-the-art and widely-used foundation model in this field. More dataset details are provided in Appendix A.

**Dataset Settings** To measure the performance of models in cross-cancer transferring, we prepare cancer-specific datasets to train specialized models. Specifically, we prepare multiple cancer-specific datasets, where each dataset corresponds to a single cancer type. For each dataset, we ensure that (*i*) its number of patients is greater than 200 and (*ii*) its ratio of event occurrence is larger than 5%, followed by splitting it into 5 folds for cross-validation-based model evaluation. The two criteria are set empirically since we find that the survival data satisfying them could often result in stable training and consistent performance across different folds. The above settings yield 13 cancer-specific datasets, which in total contain 7,268 patients with 8,818 WSIs and are used for cancer-specific training. We call them **UNI2-h-DSS-C** and denote its cancer types by $\mathcal{C} = \{c_1, \cdots, c_n\}$ with $n = 13$. The remaining 13 cancers, covering 1,922 patients with 2,370 WSIs, are cast as rare tumor diseases in TCGA (as the vast majority of them have no more than 200 patients) and are employed as test data to measure the transferability of prognostic knowledge to rare tumors. We call them **UNI2-h-DSS-R** and write its cancer types as $\mathcal{R} = \{r_1, \cdots, r_m\}$ where $m = 13$. We show the statistics of all datasets in Figure 1(b) and 1(c).

## 4 CAN WSI-BASED PROGNOSTIC KNOWLEDGE BE TRANSFERRED ACROSS DIFFERENT CANCERS?

Knowledge transfer has not yet been systemically studied in WSI-based cancer prognosis, with several issues to be investigated. In this section, we first investigate a primary question: can WSI-based prognostic knowledge be transferred across different cancer types? To answer it, we evaluate the transferability of prognostic knowledge by training cancer-specific models and measuring their performance in cross-cancer transferring. We describe the details as follows.

Concretely, transferability evaluation contains two steps. **(1) Cancer-specific training**. We follow common practices (Chen et al., 2021; Song et al., 2024b) in this step to train and evaluate WSI-based, cancer-specific prognostic models. For any cancer $c \in \mathcal{C}$, we denote its model by $\mathcal{M}_c$; it is implemented by ABMIL (Ilse et al., 2018), the most representative MIL network for WSI analysis in this field. We adopt five-fold cross-validation for performance evaluation. C-Index, as a frequently-adopted ranking-based metric in survival analysis, is reported. **(2) Model transfer**. For a specific source cancer $\mathcal{S} \in \mathcal{C}$, we transfer its tailored model ($\mathcal{M}_\mathcal{S}$) to a target cancer ($\mathcal{T}$) and measure the prognosis performance of this transferred model (denoted by $\mathcal{M}_{\mathcal{S} \to \mathcal{T}}$) on the test data of $\mathcal{T}$. Since $\mathcal{M}_\mathcal{S}$ is a special parameterization of prognostic knowledge on $\mathcal{S}$, we can quantify the knowledge transferability from $\mathcal{S}$ to $\mathcal{T}$ computationally by evaluating the performance of $\mathcal{M}_\mathcal{S}$ on $\mathcal{T}$. A transfer of $\mathcal{S} \to \mathcal{T}$ is taken as a *positive transfer* if its transfer performance (denoted by $P_{\mathcal{S} \to \mathcal{T}}$) is better than random guess, *i.e.*, $P_{\mathcal{S} \to \mathcal{T}} > 0.5$. Through setting $\mathcal{T} \in \mathcal{C}$ and $\mathcal{T} \in \mathcal{R}$, we evaluate the transferability between common cancers and to rare tumors, respectively.

We show the result of $\mathcal{T} \in \mathcal{C}$ and $\mathcal{T} \in \mathcal{R}$ in Figure 2. The diagonal metrics in Figure 2(a) show the performance of standard cancer-specific models (*i.e.*, non-transfer models). There are two notable findings from these transfer results. **(1) Negative transfer**: it is observed between several cancer diseases, as shown in Figure 2(a). Especially for the target SARC (sarcoma) and SKCM (skin cutaneous melanoma), most source models trained on other cancers cannot generalize well to them. This could be caused by intra-task and inter-task factors, *e.g.*, SARC prognosis is difficult relatively (a C-Index of 0.531 obtained by $\mathcal{M}_\mathcal{T}$), and a large data distribution gap may exist between SARC and other cancers. We will delve into these factors and elaborate on them in Section 5.2. **(2) Positive transfer**: for any $\mathcal{T} \in \mathcal{C}$, there always exists at least one positive transfer. This suggests a certain

degree of transferability across a range of cancer diseases. Further investigations are provided in Section 5.1. **(3) Transfer to rare tumors**: for all available $\mathcal{M}_\mathcal{S}$ where $\mathcal{S} \in \mathcal{C}$, we transfer them to $\mathcal{T} \in \mathcal{R}$ and measure their performance accordingly, producing the results shown in Figure 2(b). We observe that, on 8 out of 13 target diseases, there is at least one $\mathcal{M}_\mathcal{S}$ that can obtain a C-Index higher than 0.6 in cross-cancer transferring. This result indicates that cross-cancer knowledge transfer may be a promising solution to the prognosis estimation of rare tumors.

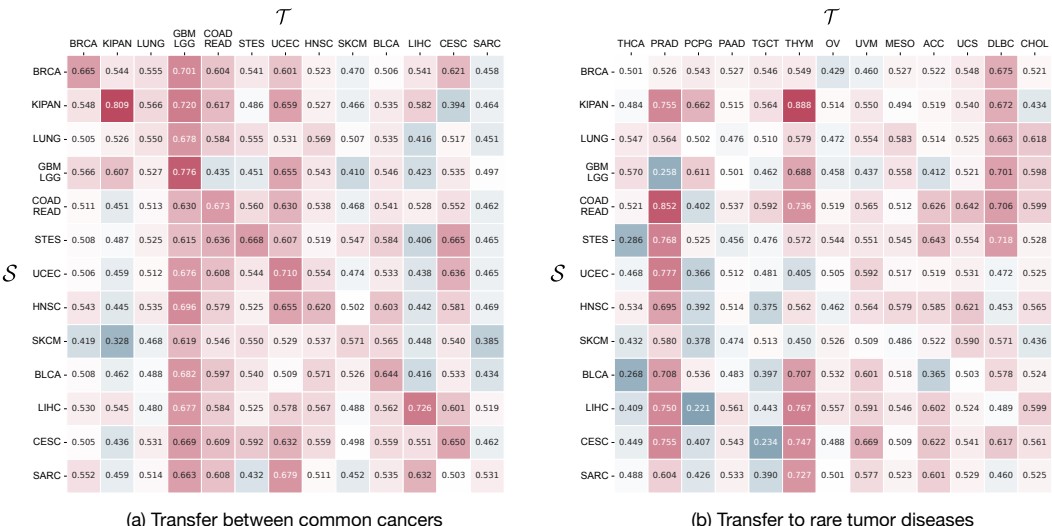

(a) Transfer between common cancers          (b) Transfer to rare tumor diseases

Figure 2: Performance of cross-cancer knowledge transfer ($\mathcal{S} \rightarrow \mathcal{T}$). Cancer-specific models are first trained on their respective datasets and are then transferred to other cancers for survival prediction. $\mathcal{M}_\mathcal{S}$ is evaluated on the five-fold test data from $\mathcal{T}$ to report transfer performance (C-Index).

## 5 WHY CAN WSI-BASED PROGNOSTIC KNOWLEDGE BE TRANSFERRED BETWEEN CERTAIN CANCERS?

Beyond a simple benchmark that merely shows the performance of cross-cancer model transfer, we further design a range of experiments to gain deeper insights into the mechanism behind transferability. Concretely, (*i*) given the positive transfer performance by $\mathcal{M}_{\mathcal{S} \rightarrow \mathcal{T}}$, we want to figure out what knowledge $\mathcal{M}_\mathcal{S}$ can offer for $\mathcal{T}$ to enable this; (*ii*) considering the result that $\mathcal{M}_\mathcal{S}$ often leads to distinct transfer performances across different $\mathcal{T}$, we want to study what factors affect cross-cancer knowledge transfer. Next, we describe experimental designs and notable insights.

### 5.1 WHAT KNOWLEDGE CAN WSI-BASED PROGNOSTIC MODELS OFFER TO ENABLE POSITIVE TRANSFER?

We investigate the features that $\mathcal{M}_{\mathcal{S} \rightarrow \mathcal{T}}$ can offer via visualizing and comparing the attention heatmaps derived from $\mathcal{M}_\mathcal{T}$ and $\mathcal{M}_{\mathcal{S} \rightarrow \mathcal{T}}$, where $\mathcal{S}$ is set to the source nearest to $\mathcal{T}$ (*i.e.*, the source that leads to the best $\mathcal{M}_{\mathcal{S} \rightarrow \mathcal{T}}$) for better view. Moreover, we show the annotation of tissue types in WSIs to examine if the areas that a transferred model focuses on are relevant to prognosis. However, manually annotating tissue types is labor-intensive for gigapixel WSIs. To address this, we choose colorectal cancer (COADREAD) as an example and utilize a vision-language model, CONCH (Lu et al., 2024), for tissue annotation, as it obtains an accuracy of 94% and demonstrates state-of-the-art performance in classifying the tissue type of colorectal cancer (Kather et al., 2019b). Refer to Appendix B.2 for more details. Consequently, all test samples from COADREAD are annotated by CONCH and are passed through $\mathcal{M}_\mathcal{T}$ and $\mathcal{M}_{\mathcal{S} \rightarrow \mathcal{T}}$ to produce attention maps.

By visualizing and comparing tissue annotations and attention heatmaps, we observe that the knowledge that prognostic models offer can be basically decoupled into three representative parts. To exhibit them, we select three test WSIs and show their results in Figure 3. Our observations are summarized as follows. **(1) Overlapping and useful regions**. As shown in Figure 3(a), $\mathcal{M}_{\mathcal{S} \rightarrow \mathcal{T}}$

can also identify the tumor and stroma tissues that contribute to prognosis estimation, just like the specialized $\mathcal{M}_{\mathcal{T}}$. A primary reason is that there are general cellular prognostic patterns across different tissues (Yu et al., 2016; Wulczyn et al., 2020; Chen et al., 2022), *e.g.*, large and irregularly shaped nuclei, unclear boundaries between tumor and normal tissue, *etc*, and the transferred $\mathcal{M}_{\mathcal{S}}$ is able to identify these patterns. **(2) Dissimilar and useless regions** in Figure 3(b). $\mathcal{M}_{\mathcal{S}\to\mathcal{T}}$ sometimes mistakenly take unrelated regions (*e.g.*, muscle) as its focus, while $\mathcal{M}_{\mathcal{T}}$ can ignore them. This could be caused by the intrinsic difference between $\mathcal{S}$ and $\mathcal{T}$ in tissue phenotype. **(3) Dissimilar yet useful regions** in Figure 3(c). Most importantly, we observe that $\mathcal{M}_{\mathcal{S}\to\mathcal{T}}$ can identify the complete and useful regions overlooked by $\mathcal{M}_{\mathcal{T}}$. This capability of $\mathcal{M}_{\mathcal{S}}$ not only contributes to a positive transfer but also could be utilized to improve the model's performance on $\mathcal{T}$.

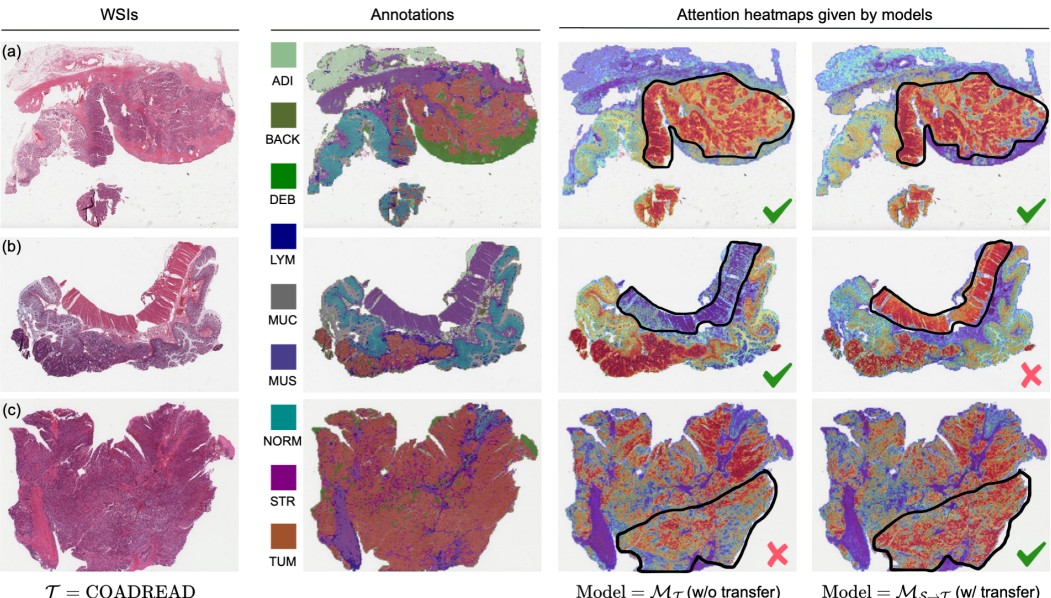

Figure 3: Visualization of tissue annotations and attention heatmaps to showcase representative knowledge that $\mathcal{M}_{\mathcal{S}\to\mathcal{T}}$ offers: (a) overlapping and useful, (b) dissimilar and useless, and (c) dissimilar yet useful, where (a) and (c) help to positive transfer. Refer to Appendix C.1 for more results.

## 5.2 WHAT FACTORS AFFECT THE TRANSFER OF WSI-BASED PROGNOSTIC KNOWLEDGE?

We posit there are two groups of factors that affect the transfer of WSI-based prognostic knowledge from $\mathcal{S}$ to $\mathcal{T}$—*intra-task* and *inter-task factors*, as the transfer involves both the tasks themselves and the correlation between tasks. To confirm it, we adopt statistical tools to analyze these factors. We elucidate these factors and the statistical results below.

**For intra-task factors**, we consider two intuitive and critical factors: (*i*) $P_{\mathcal{S}}$, measuring the goodness of $\mathcal{M}_{\mathcal{S}}$, as $\mathcal{M}_{\mathcal{S}}$ is more likely to perform well on $\mathcal{T}$ if it is good enough in its own domain; (*ii*) $P_{\mathcal{T}}$, reflecting the difficulty of accurately estimating the prognosis of $\mathcal{T}$, because $\mathcal{M}_{\mathcal{S}}$ likely fails on $\mathcal{T}$ if the prognosis of $\mathcal{T}$ is complicated and difficult to predict based on WSIs. **For inter-task factors**, we consider one oncology-relevant factor and one common factor in knowledge transfer: (*i*) $C_{\mathcal{S}\to\mathcal{T}}^{\text{RMST}}$, indicating the closeness of $\mathcal{S}$ to $\mathcal{T}$ in tumor invasiveness; (*ii*) $C_{\mathcal{S}\to\mathcal{T}}^{\text{Dist.}}$, the closeness of $\mathcal{S}$ to $\mathcal{T}$ in data distribution. We measure tumor invasiveness using the restricted mean survival time (RMST) within 10 years, as RMST is a metric assessing overall survival, and a longer RMST implies weaker invasiveness. The distance from $\mathcal{S}$ to $\mathcal{T}$ in data distribution is measured by the Euclidean distance between the centroid of mean-based WSI embeddings in $\mathcal{S}$ and $\mathcal{T}$.

We conduct univariate and multivariate statistical analysis for the above four factors. Beta regression (Ferrari & Cribari-Neto, 2004) is adopted for multivariate analysis as the response variable C-Index is bounded in [0, 1] and with heteroskedasticity. We show results in Figure 4 and Table 1 and analyze them as follows. **(1)** $P_{\mathcal{S}}$: In univariate analysis, we examine the correlation of $P_{\mathcal{S}}$ with the overall performance of transferring $\mathcal{M}_{\mathcal{S}}$, *i.e.*, the average over $\{P_{\mathcal{S}\to\mathcal{T}}|\mathcal{T}\in\mathcal{C}\}$.

From the results shown in Figure 4(a), we find that $P_S$ has a positive correlation (Pearson correlation = 0.508). However, it is not significant in Beta regression. This indicates that there are other important factors. **(2)** $P_T$: In the univariate analysis shown in Figure 4(b), we quantify its relation with the overall performance of transferring different models to $T$, *i.e.*, the average over $\{P_{S\to T}|S\in\mathcal{C}\}$, obtaining the similar statistical results to those of $P_S$. Besides, multivariate analysis shows that $P_T$ is a significant factor. **(3)** $C^{RMST}_{S\to T}$: The statistical result in Table 1 shows that it helps to explain the variation to some extent, though not significantly. In univariate analysis, we find significant correla-

Table 1: Multivariate analysis with Beta regression. It assesses whether factors can explain the variations in transfer performance. $\phi$ is the precision of Beta distribution. NLL = Negative Log Likelihood. * P-value $\leq 0.05$.

| Factors | | | | Beta regression | | |
|---|---|---|---|---|---|---|
| $P_S$ | $P_T$ | $C^{RMST}_{S\to T}$ | $C^{Dist.}_{S\to T}$ | $\phi$ (↑) | NLL (↓) | P-value |
| ✓ | | | | 3.617 | -186.2 | |
| | ✓ | | | 3.726 | -195.3 | * |
| ✓ | ✓ | | | 3.742 | -196.7 | /* |
| ✓ | ✓ | ✓ | | 3.745 | -196.9 | /*/ |
| ✓ | ✓ | ✓ | ✓ | 3.866 | -207.2 | */*/* |
| ✓ | ✓ | ✓ | ✓ | **3.883** | **-208.6** | */*/ /* |

tions between $C^{RMST}_{S\to T}$ and $P_{S\to T}-P_S$ (the performance increase in transferring $\mathcal{M}_S$ to $T$) on target cancers like BLCA, as shown in Figure 4(c). Note that $P_S$ is subtracted to exclude the impact of $P_S$. **(4)** $C^{Dist.}_{S\to T}$: Beta regression shows that it is a critical factor affecting knowledge transfer (P-value $< 0.05$). In addition, a significant correlation between $C^{Dist.}_{S\to T}$ and $P_{S\to T}-P_S$ is also identified on BLCA, as shown in Figure 4(d). **(5)** Finally, combining all factors yields the best performance in Table 1. These results suggest that the four defined factors help explain cross-cancer transfer performance with varying degrees of contribution. Additional results, *e.g.*, the univariate analysis of inter-task factors on other cancers, are provided in Appendix C.2.

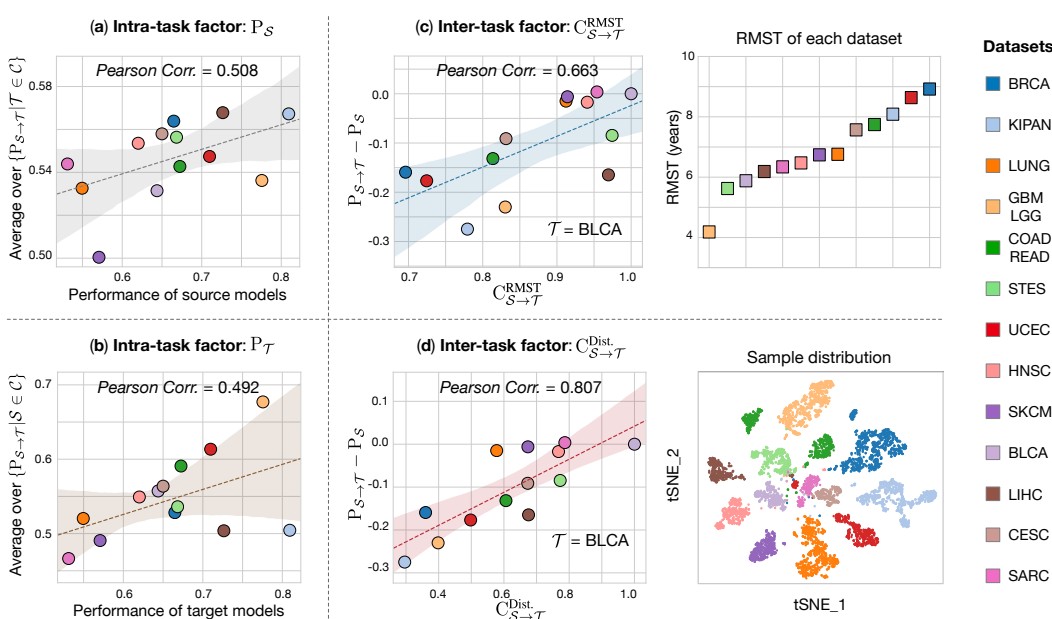

Figure 4: Univariate analysis and visualization for intra-task and inter-task factors.

# 6 CAN CROSS-CANCER KNOWLEDGE TRANSFER FURTHER IMPROVE PROGNOSTIC PERFORMANCE?

The above studies have demonstrated the transferability of WSI-based prognostic knowledge in Section 4 and provided notable insights into its underlying mechanism in Section 5. Here, we aim to investigate *the utility of cross-cancer knowledge transfer*, *i.e.*, can we utilize transferred knowledge to improve prognostic performance further? To this end, we propose a baseline approach to showcase that the knowledge transferred from other cancers can be effectively leveraged to improve model performance. Next, we present its details, along with experimental results and analysis.

## 6.1 A Routing-Based Baseline Approach to Utilizing Knowledge Transferred from Other Cancers

Given multiple off-the-shelf models $\{\mathcal{M}_\mathcal{S} \mid \mathcal{S} \in \mathcal{C}\}$, it is anticipated to make use of their beneficial and generalizable knowledge to improve the model's performance on $\mathcal{T}$. However, for a given target $\mathcal{T} \in \mathcal{C}$, it is unclear which $\mathcal{M}_\mathcal{S}$ could offer such helpful information. Moreover, one $\mathcal{M}_\mathcal{S}$ could offer useful knowledge for some inputs, but misleading cues for others, as suggested by Figure 3. Thus, we propose a routing-based baseline approach, named ROUPKT, inspired by the core idea behind Mixture of Experts (MoE) (Jacobs et al., 1991; Shazeer et al., 2017). In a nutshell, we propose to use the routing mechanism to adaptively select useful knowledge combinations from all available $\mathcal{M}_\mathcal{S}$. We introduce the architecture and implementation of ROUPKT below.

**Architecture**: For any target $\mathcal{T}$, in ROUPKT, all off-the-shelf models, *i.e.*, $\mathcal{M}_\mathcal{T}$ and $\{\mathcal{M}_\mathcal{S} \mid \mathcal{S} \in \mathcal{C} \setminus \{\mathcal{T}\}\}$, are treated as prognostic experts, denoted by $\{\boldsymbol{\mathcal{E}}_\tau\}_{\tau=1}^n$. As shown in figure 5, a router is placed to adaptively select the optimal experts that can offer complementary knowledge to enable better prognosis prediction for different WSI inputs. Concretely, a preprocessed gigapixel WSI, denoted by a bag of instances $\boldsymbol{X} = \{x_i\}_{i=1}^M \in \mathbb{R}^{M \times d}$, is taken as input. It is first passed through a router to produce the routing scores of all experts, $\{w_\tau\}_{\tau=1}^n$. Then, based on $\{w_\tau\}_{\tau=1}^n$, top $K$ experts are selected and their corresponding outputs (WSI-level features) are aggregated into a mixed representation for prediction.

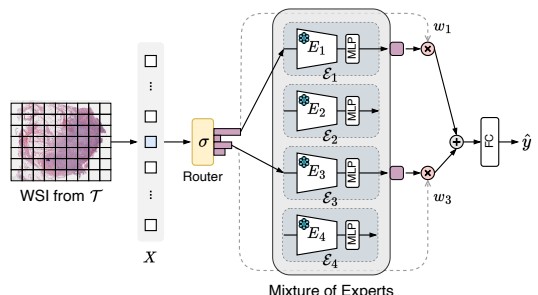

Figure 5: A schematic diagram of our routing-based baseline approach (ROUPKT) to utilizing the knowledge transferred from other cancers.

**Implementation**: In ROUPKT, the router is a classical attention-based MIL network that first processes the aggregation of multi-instances and then outputs $\{w_\tau\}_{\tau=1}^n$ for expert routing. $K$ is set to 5 by default. An expert $\boldsymbol{\mathcal{E}}_\tau$ consists of *(i)* a frozen MIL encoder $\boldsymbol{E}_\tau$ taken from a transferred, fitted model $\mathcal{M}_\tau$ and *(ii)* a trainable MLP (multi-layer perceptron) used to adapt the fixed $\boldsymbol{E}_\tau(\boldsymbol{X})$ to the target task. The prediction head is implemented by a FC (fully-connected) layer. Note that $\boldsymbol{E}_\tau$ is frozen, so $\{\boldsymbol{E}_\tau(\boldsymbol{X})\}_{\tau=1}^n$ are fixed and can be computed in advance to maintain training efficiency. More details regarding implementation and model training can be found in Appendix B.5.

### 6.2 Experimental Results and Analysis

① **Prognostic knowledge can be effectively transferred across cancers to improve model performance via ROUPKT**: To verify whether the model can benefit from cross-cancer knowledge transfer, we compare ROUPKT with $\mathcal{M}_\mathcal{T}$ and other common models: $\mathcal{M}_{\mathcal{S} \to \mathcal{T}}$, $\boldsymbol{E}_\mathcal{T}$ + MLP, and $\boldsymbol{E}_{\mathcal{S} \to \mathcal{T}}$ + MLP. The last two are fine-tuned models, where an MLP is trained for prediction, and the encoder $\boldsymbol{E}$ is either frozen or free in fine-tuning. $\mathcal{S}$ is set to the source with the best performance on the validation set of $\mathcal{T}$. Comparative results are exhibited in Table 2. We observe that *(i)* ROUPKT obtains the best performance in 9 out of 13 target cancers and *(ii)* most importantly, it achieves the best average performance with an improvement of 3.1% over $\mathcal{M}_\mathcal{T}$. These notable results confirm that, by using ROUPKT, the WSI-based prognostic knowledge from other cancers can often be effectively utilized to improve model performance on the target cancer. This directly reflects the utility of cross-cancer knowledge transfer in WSI-based prognosis prediction.

② **Expert routing is a superior strategy for utilizing various transferred knowledge**: Expert routing is the core of ROUPKT. It can adaptively combine the most beneficial knowledge transferred from $\mathcal{M}_\mathcal{S}$. To verify its advantage, we compare it with well-known strategies that can utilize and combine multiple experts $\{\mathcal{M}_\mathcal{S} \mid \mathcal{S} \in \mathcal{C}\}$. They contain *(i)* classical MIL aggregation methods (MeanMIL and attention in ABMIL), *(ii)* model ensemble that averages over the survival prediction of all experts, and *(iii)* representative sequence-based learning methods like RNN in GRU (Cho et al., 2014) and self-attention in Transformer (Vaswani et al., 2017). The only difference between them and ROUPKT is the way of processing multiple experts. From the results shown in Table 3,

Table 2: Comparison with $\mathcal{M}_\mathcal{T}$ and other fine-tuned or transferred models. $^\dagger$ The encoder $\boldsymbol{E}$ of these baselines is involved in fine-tuning, not in a frozen state.

| Model | Performance on $\mathcal{T}$ | | | | | | | | | | | | | Avg. |
|---|---|---|---|---|---|---|---|---|---|---|---|---|---|---|
| | BRCA | KIPAN | LUNG | GBM LGG | COAD READ | STES | UCEC | HNSC | SKCM | BLCA | LIHC | CESC | SARC | |
| $\mathcal{M}_\mathcal{T}$ | 0.6648 (± 0.032) | 0.8094 (± 0.013) | 0.5496 (± 0.034) | 0.7756 (± 0.020) | 0.6725 (± 0.028) | 0.6648 (± 0.040) | 0.7098 (± 0.043) | 0.6201 (± 0.047) | 0.5708 (± 0.037) | 0.6438 (± 0.015) | 0.7265 (± 0.048) | 0.6500 (± 0.058) | 0.5312 (± 0.056) | 0.6609 |
| $\boldsymbol{E}_\mathcal{T}$ + MLP | 0.6723 (± 0.021) | 0.8080 (± 0.014) | 0.5463 (± 0.034) | 0.7744 (± 0.020) | 0.6753 (± 0.028) | 0.6694 (± 0.044) | 0.7136 (± 0.039) | 0.6207 (± 0.043) | 0.5682 (± 0.032) | 0.6384 (± 0.021) | 0.7253 (± 0.046) | 0.6367 (± 0.058) | 0.5462 (± 0.030) | 0.6611 |
| $\boldsymbol{E}_\mathcal{T}$ + MLP $^\dagger$ | 0.6304 (± 0.053) | 0.7972 (± 0.025) | 0.5757 (± 0.052) | 0.7657 (± 0.016) | 0.6663 (± 0.039) | 0.6761 (± 0.051) | 0.6880 (± 0.059) | 0.6214 (± 0.045) | 0.5857 (± 0.035) | 0.6211 (± 0.037) | 0.7209 (± 0.043) | 0.6440 (± 0.086) | 0.5034 (± 0.062) | 0.6535 |
| $\mathcal{M}_{\mathcal{S}\to\mathcal{T}}$ | 0.5450 (± 0.050) | 0.6035 (± 0.031) | 0.5692 (± 0.052) | 0.7130 (± 0.033) | 0.6417 (± 0.023) | 0.6096 (± 0.041) | 0.6426 (± 0.069) | 0.5649 (± 0.052) | 0.5422 (± 0.029) | 0.6205 (± 0.043) | 0.5441 (± 0.058) | 0.6712 (± 0.046) | 0.5093 (± 0.052) | 0.5982 |
| $\boldsymbol{E}_{\mathcal{S}\to\mathcal{T}}$ + MLP | 0.5288 (± 0.058) | 0.6430 (± 0.050) | 0.5406 (± 0.035) | 0.7713 (± 0.022) | 0.6345 (± 0.039) | 0.6155 (± 0.042) | 0.6658 (± 0.086) | 0.5759 (± 0.065) | 0.5206 (± 0.036) | 0.5971 (± 0.024) | 0.6023 (± 0.067) | 0.6177 (± 0.046) | 0.4764 (± 0.051) | 0.5992 |
| $\boldsymbol{E}_{\mathcal{S}\to\mathcal{T}}$ + MLP $^\dagger$ | 0.6309 (± 0.054) | **0.8105** (± 0.020) | **0.5872** (± 0.071) | **0.7771** (± 0.020) | 0.6705 (± 0.019) | 0.6617 (± 0.044) | 0.6843 (± 0.058) | 0.6198 (± 0.036) | 0.5723 (± 0.025) | 0.6356 (± 0.041) | 0.7061 (± 0.072) | **0.6785** (± 0.075) | 0.4936 (± 0.059) | 0.6560 |
| **ROUPKT** | **0.7181** (± 0.051) | 0.8096 (± 0.007) | 0.5714 (± 0.033) | 0.7726 (± 0.019) | **0.7123** (± 0.047) | **0.6708** (± 0.042) | **0.7371** (± 0.054) | **0.6257** (± 0.034) | **0.5954** (± 0.025) | **0.6644** (± 0.017) | **0.7563** (± 0.048) | 0.6629 (± 0.061) | **0.5596** (± 0.040) | **0.6812** |

we observe that the routing in ROUPKT obtains the best overall performance, and it often outperforms other strategies by a large margin. These results suggest that routing is a superior approach to utilizing the WSI-based prognostic knowledge transferred from other cancers.

Table 3: Comparison with the other strategies that can utilize and combine multiple experts.

| Strategy | Performance on $\mathcal{T}$ | | | | | | | | | | | | | Avg. |
|---|---|---|---|---|---|---|---|---|---|---|---|---|---|---|
| | BRCA | KIPAN | LUNG | GBM LGG | COAD READ | STES | UCEC | HNSC | SKCM | BLCA | LIHC | CESC | SARC | |
| Mean | 0.5708 (± 0.053) | 0.7273 (± 0.031) | 0.5236 (± 0.041) | 0.7753 (± 0.020) | 0.6227 (± 0.032) | 0.6076 (± 0.049) | 0.7122 (± 0.084) | 0.5578 (± 0.067) | 0.5001 (± 0.026) | 0.6063 (± 0.032) | 0.5901 (± 0.058) | 0.6075 (± 0.058) | 0.4322 (± 0.063) | 0.6026 |
| Ensemble | 0.5972 (± 0.045) | 0.7804 (± 0.012) | 0.5322 (± 0.018) | 0.7753 (± 0.041) | 0.6287 (± 0.050) | 0.6146 (± 0.089) | 0.7160 (± 0.063) | 0.5631 (± 0.040) | 0.4976 (± 0.024) | 0.6191 (± 0.030) | 0.6380 (± 0.048) | 0.5987 (± 0.055) | 0.4386 (± 0.065) | 0.6153 |
| Attention (in ABMIL) | 0.6729 (± 0.024) | 0.8079 (± 0.011) | 0.5478 (± 0.034) | 0.7737 (± 0.021) | 0.6694 (± 0.034) | 0.6582 (± 0.050) | 0.7089 (± 0.033) | 0.5770 (± 0.057) | 0.5503 (± 0.045) | 0.5917 (± 0.041) | 0.7093 (± 0.037) | 0.6366 (± 0.020) | 0.4772 (± 0.058) | 0.6447 |
| Gated-Attention (in ABMIL) | 0.6772 (± 0.025) | 0.8077 (± 0.012) | 0.5426 (± 0.032) | 0.7744 (± 0.021) | 0.6701 (± 0.035) | 0.6458 (± 0.048) | 0.7104 (± 0.032) | 0.5714 (± 0.065) | 0.5479 (± 0.048) | 0.5754 (± 0.045) | 0.7031 (± 0.045) | 0.6401 (± 0.019) | 0.4527 (± 0.054) | 0.6399 |
| RNN (in GRU) | 0.5577 (± 0.066) | 0.7715 (± 0.013) | 0.5245 (± 0.041) | **0.7828** (± 0.020) | 0.6129 (± 0.048) | 0.5829 (± 0.041) | 0.7334 (± 0.065) | 0.5492 (± 0.086) | 0.4786 (± 0.028) | 0.5486 (± 0.020) | 0.6624 (± 0.083) | 0.6448 (± 0.081) | 0.4893 (± 0.070) | 0.6106 |
| Self-Attention (in Transformer) | 0.5828 (± 0.033) | 0.7992 (± 0.008) | 0.5411 (± 0.033) | 0.7645 (± 0.015) | 0.6745 (± 0.032) | 0.6682 (± 0.040) | 0.7150 (± 0.048) | 0.6237 (± 0.045) | 0.5440 (± 0.038) | 0.5904 (± 0.024) | 0.7316 (± 0.059) | 0.6400 (± 0.054) | 0.5256 (± 0.018) | 0.6462 |
| **Routing (in ROUPKT)** | **0.7181** (± 0.051) | **0.8096** (± 0.007) | **0.5714** (± 0.033) | 0.7726 (± 0.019) | **0.7123** (± 0.047) | **0.6708** (± 0.042) | **0.7371** (± 0.054) | **0.6257** (± 0.034) | **0.5954** (± 0.025) | **0.6644** (± 0.017) | **0.7563** (± 0.048) | **0.6629** (± 0.061) | **0.5596** (± 0.040) | **0.6812** |

**Additional experiments**: We further conduct ablation studies on ROUPKT to examine the impact of different expert combinations. Their results are shown in Appendix D.1. Additional experiments and analysis, such as case studies and visualization for expert routing, hyper-parameter analysis, comparisons with other MIL networks, *etc*, are provided in Appendix D.

## 7 CONCLUSION

This paper presents CROPKT, the first preliminary yet systematic study on cross-cancer prognosis knowledge transfer in WSIs. Firstly, we curate a large WSI dataset with 26 cancer types, UNI2-h-DSS, to quantify the transferability of WSI-based prognostic knowledge across different cancers, including the transferability to rare tumors. Secondly, to gain a deeper understanding of the mechanism behind transferability, we further design a range of experiments. With attention maps and statistical methods, we find that a model transferred from another cancer can offer useful knowledge overlooked by the model on a target cancer, and there are four intra-task and inter-task factors affecting transferability. Finally, to demonstrate the utility of cross-cancer knowledge transfer in WSI-based prognosis, we propose a routing-based baseline approach, ROUPKT. Empirical experiments confirm that, with ROUPKT, cross-cancer knowledge can be effectively leveraged to improve prognosis performance by 3.1% on average, and routing is a superior mechanism in utilizing multiple transferred, off-the-shelf prognostic models. We hope CROPKT could lay the foundation and inspire more studies on WSI-based prognosis with cross-cancer knowledge transfer.

ETHICS STATEMENT

Our research focuses on studying cross-cancer knowledge transfer in WSI-based prognosis pre-diction, involving the evaluation, insights, and utility of the transferability. All experiments are performed on widely used, publicly available benchmark datasets (*i.e.*, TCGA). We did not interact with human subjects or create new human datasets. The work poses no foreseeable negative societal impact and aligns with the ICLR Code of Ethics.

REPRODUCIBILITY STATEMENT

To facilitate the reproducibility of our work, we commit to the following: (1) for the dataset used in the experiments, a complete description of data processing steps is provided in Appendix; (2) for the implementation of the proposed algorithms and the relevant experiments, a zip file with anonymous source code is submitted as supplementary materials. Additionally, our dataset, code, and models will be made publicly available in the future for reproducibility.

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

APPENDIX

# A    DATASET DETAILS

## A.1    WSI DATASETS

It contains 11,646 WSIs from TCGA[2]. All WSIs have been processed into patch features[3] by UNI2-h (Chen et al., 2024): for each WSI, tissue regions are first segmented and then tiled into small image patches with $256 \times 256$ pixels at $20\times$ magnification, followed by extracting image features (with dimensionality of 1,536) from each patch with UNI2-h. UNI2-h is trained on private WSI data (not TCGA) and has been widely adopted as a patch feature extractor for WSIs. In the beginning, 11,646 WSI samples cover 32 TCGA projects, with some different projects belonging to the same organ. According to the conventions in this field, we merge the following projects into a larger one: (*i*) TCGA-KICH, TCGA-KIRC, and TCGA-KIRP into KIPAN, (*ii*) TCGA-LUSC and TCGA-LUAD into LUNG, (*iii*) TCGA-ESCA and TCGA-STAD into STES, (*iv*) TCGA-GBM and TCGA-LGG into GBMLGG, (*v*) TCGA-COAD and TCGA-READ into COADREAD. This results in 26 cancer types, as shown in Figure 1.

## A.2    SETTINGS OF UNI2-H-DSS-C

UNI2-h-DSS-C is curated to train and evaluate cancer-specific prognostic models. It contains 13 cancer-specific cohorts. Their statistics are provided in Table 4. We split each cohort into five folds and apply five-fold cross-validation for performance evaluation. Continuous DSS follow-up time is converted into discrete, with year as time unit. Specially, in data split, we perform patient-level splitting, stratified by the number of censored and uncensored patients in each discrete time bin to ensure the balance of censorship rate in each fold. As the balance is hard to obtain on the cohort with limited samples (*e.g.*, $N < 300$), we repeat the splitting using different random seeds until one certain split can obtain a relatively stable performance (namely, a small variance across folds) in five-fold cross-validation. We do this to ensure a fair benchmark evaluation.

Table 4: Statistics of datasets in UNI2-h-DSS-C. Patch number is calculated at patient level.

| Dataset | # Patients | Survival rate | # WSIs | # Patches (avg.) | # Patches (max) | Storage |
|---------|-----------|---------------|--------|------------------|-----------------|---------|
| BRCA | 1,035 | 92.4% | 1,106 | 10,694 | 58,176 | 65 GB |
| KIPAN | 880 | 83.6% | 910 | 10,117 | 72,795 | 52 GB |
| LUNG | 848 | 78.2% | 924 | 13,199 | 220,294 | 72 GB |
| GBMLGG | 841 | 52.4% | 1,623 | 16,209 | 216,885 | 82 GB |
| COADREAD | 568 | 86.6% | 577 | 6,458 | 38,322 | 22 GB |
| STES | 512 | 72.7% | 539 | 8,429 | 31,458 | 26 GB |
| UCEC | 504 | 89.3% | 565 | 14,203 | 73,141 | 42 GB |
| HNSC | 427 | 71.7% | 449 | 6,530 | 56,558 | 17 GB |
| SKCM | 412 | 60.7% | 453 | 9,593 | 136,934 | 24 GB |
| BLCA | 372 | 68.5% | 443 | 14,465 | 103,807 | 32 GB |
| LIHC | 355 | 77.5% | 362 | 8,875 | 35,839 | 19 GB |
| CESC | 266 | 81.6% | 276 | 6,212 | 54,021 | 10 GB |
| SARC | 248 | 68.5% | 591 | 27,346 | 243,906 | 40 GB |
| All | 7,268 | 76.8% | 8,818 | 11,509 | 243,906 | 503 GB |

## A.3    DATA ACCESSIBILITY

All datasets, including UNI2-h-DSS-C and UNI2-h-DSS-R, along with complete follow-up labels and data split files, will be made publicly-available here[4].

---

[2]https://portal.gdc.cancer.gov/

[3]https://huggingface.co/datasets/MahmoodLab/UNI-h-features

[4]https://huggingface.co/datasets

## B  IMPLEMENTATION DETAILS

### B.1  CANCER-SPECIFIC TRAINING

For any specific cancer $c \in \mathcal{C}$, $\mathcal{M}_c$ is a standard ABMIL network. Its network implementation and training recipe follow Song et al. (2024b). (1) Specifically, for **network implementation**, ABMIL is composed of an MLP as an instance embedding layer, a gated attention layer for multi-instance aggregation, and a fully-connected layer as a prediction head. (2) For **network training**, $\mathcal{M}_c$ adopts a learning rate of 0.0001, an optimizer of AdamW with a weight decay of 0.00001, and a batch size of 1 (one bag) with 16 steps for gradient accumulation, trained for 20 epochs using a NLL loss frequently-adopted in survival analysis (Zadeh & Schmid, 2020). Besides, learning rate is adjusted by a cosine annealing schedule, where the epoch number of warming up is set to 1.

### B.2  TISSUE ANNOTATION

CONCH (Lu et al., 2024) is trained to align visual features with textual features in a latent space. Thus, we adopt it to classify image patches into different tissue types by measuring the similarity between each image patch and the text prompts describing tissue types. We adopt the text prompts same as CONCH and follow CONCH to use MI-Zero (Lu et al., 2023) for tissue classification. More implementation details can be found here[5].

### B.3  CALCULATION OF INTER-TASK FACTORS

**For** $C_{\mathcal{S} \rightarrow \mathcal{T}}^{\mathbf{RMST}}$, we first calculate the RMST (restricted mean survival time) within 10 years for $c \in \mathcal{C}$, as shown in Figure 4(c), according to its definition:

$$\mathrm{RMST}_c(t) = \int_0^t S_c(\tau)d\tau,$$

where $S_c$ is the survival function of the population with $c$. From the above equation, we can find that RMST is the average survival time during a defined time period (Kim et al., 2017). When patients have a cancer disease with stronger invasiveness, their overall survival time would be shorter, leading to a smaller RMST. Thus, RMST is adopted in this study to reflect the overall invasiveness of cancer. Then, we normalize RMST and measure the closeness of $\mathcal{S} \rightarrow \mathcal{T}$ in terms of RMST by

$$C_{\mathcal{S} \rightarrow \mathcal{T}}^{\mathbf{RMST}} = 1 - \frac{|\mathrm{RMST}_{\mathcal{S}}(10) - \mathrm{RMST}_{\mathcal{T}}(10)|}{10}.$$

**For** $C_{\mathcal{S} \rightarrow \mathcal{T}}^{\mathbf{Dist.}}$, we project all slide-level features of 13 cancer datasets into a 2D plane using t-SNE, as shown in Figure 4(d). Then, we measure the distance between $\mathcal{S}$ and $\mathcal{T}$ (denoted as $D_{\mathcal{S},\mathcal{T}}$) by simply calculating the L2 distance between their respective central embeddings, where the average of all slide features is taken as the central embedding of one cancer-specific dataset. Finally, we normalize it and measure the closeness of $\mathcal{S} \rightarrow \mathcal{T}$ in terms of distribution by

$$C_{\mathcal{S} \rightarrow \mathcal{T}}^{\mathbf{Dist.}} = 1 - \frac{D_{\mathcal{S},\mathcal{T}}}{\max(D)}.$$

### B.4  OLS REGRESSION

Multivariate analysis is conducted using OLS regression. It outputs (*i*) an adjusted $R^2$ that measures how much variations of dependent variable can be explained by predictors and (*ii*) a NLL (negative log likelihood) that measures the goodness of fit. Moreover, the significance of each predictor can be assessed accordingly. A Python package, statsmodels[6], is adopted for these.

### B.5  OUR ROUTING-BASED BASELINE APPROACH

**Network Settings**  For any target task $\mathcal{T} \in \mathcal{C}$, we utilize all available MIL encoders $\{\boldsymbol{E}_{\mathcal{S}} \mid \mathcal{S} \in \mathcal{C}\}$ from $\{\mathcal{M}_{\mathcal{S}} \mid \mathcal{S} \in \mathcal{C}\}$ to construct MoE. They are set to be frozen in training. In the training phrase

---

[5]https://github.com/mahmoodlab/CONCH
[6]https://www.statsmodels.org

of the $i$-th fold, we only use the $\boldsymbol{E}_{\mathcal{S}=\mathcal{T}}$ obtained from the same fold to prevent data leakage; the other MIL encoders, $\{\boldsymbol{E}_{\mathcal{S}} \mid \mathcal{S} \in \mathcal{C} \setminus \{\mathcal{T}\}\}$, are from the first fold by default. $K$ is set to 5 by default without hyper-parameter tuning. Additional experiments regarding hyper-parameter analysis are given in Appendix D.2.

**Model Training** It remains the same as that in cancer-specific training unless otherwise specified. Specially, to stabilize the training of MoE in ROUPKT, we follow common practices (Zoph et al., 2022) to adopt two auxiliary objectives: $\mathcal{L}_B$ and $\mathcal{L}_Z$. $\mathcal{L}_B$ is used to balance the load of each expert. We observe a small coefficient for it is often better, so we set it to 0.01 by default without tuning. $\mathcal{L}_Z$ is used to penalize the very large scores in routing. We set it to 0.01 after a grid search over $\{0, 0.001, 0.005, 0.01\}$ using the train set of the first three datasets (*i.e.*, BRCA, KIPAN, and LUNG). Readers could refer to Zoph et al. (2022) for the details of $\mathcal{L}_B$ and $\mathcal{L}_Z$. Routing noise is not used as in Zoph et al. (2022) since we observe it often leads to unstable performance. All experiments are run on a machine with two NVIDIA GeForce RTX 3090 GPUs (24G).

**Source Code** Please refer to our supplementary files. We will also make it publicly available for reproducibility.

## C  ADDITIONAL RESULTS OF TRANSFERABILITY INSIGHTS

### C.1  MORE EXAMPLES OF DISSIMILAR YET USEFUL REGIONS

Here we give more representative examples in Figure 6, in addition to that shown in Figure 3. All of exhibited WSI samples are from test set. For the first example ($\mathcal{T} = $ BRCA) and the third one ($\mathcal{T} = $ LUNG), we observe that $\mathcal{M}_{\mathcal{S} \to \mathcal{T}}$ captures the regions that are helpful for prognosis prediction but are overlooked by $\mathcal{M}_{\mathcal{T}}$. For the second one ($\mathcal{T} = $ BLCA), $\mathcal{M}_{\mathcal{T}}$ shows very high attention scores on useless regions, while they can be successfully ignored by $\mathcal{M}_{\mathcal{S} \to \mathcal{T}}$.

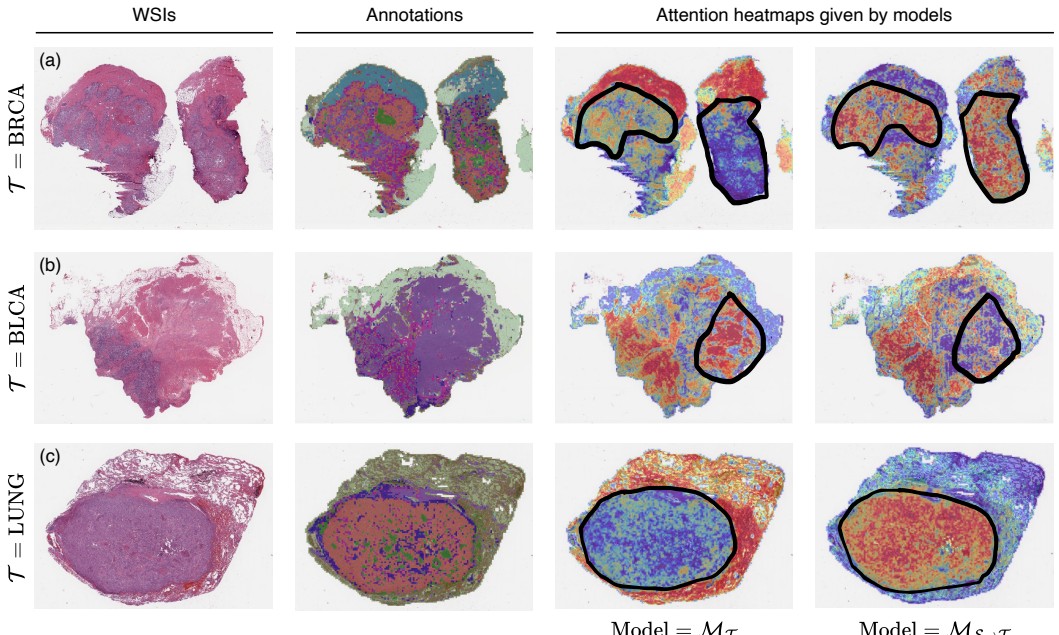

Figure 6: More examples of dissimilar yet useful regions.

### C.2  MORE RESULTS ON INTER-TASK FACTORS

**Univariate Analysis** On the other cancers beyond BLCA (shown in Figure 4(c) and 4(d)), we also find significant correlations (P-value $< 0.05$) between inter-task factors and the performance increase in transferring $\mathcal{M}_{\mathcal{S}}$ to $\mathcal{T}$ ($P_{\mathcal{S} \to \mathcal{T}} - P_{\mathcal{S}}$). Their results are shown in Figure 7(a) and 7(b)

for $C_{S\to T}^{\text{RMST}}$ and $C_{S\to T}^{\text{Dist.}}$, respectively. On the remaining target cancers, only positive correlations are observed. Even so, both two inter-task factors can explain the variations of $P_{S\to T}$, as indicated by the results in Table 1.

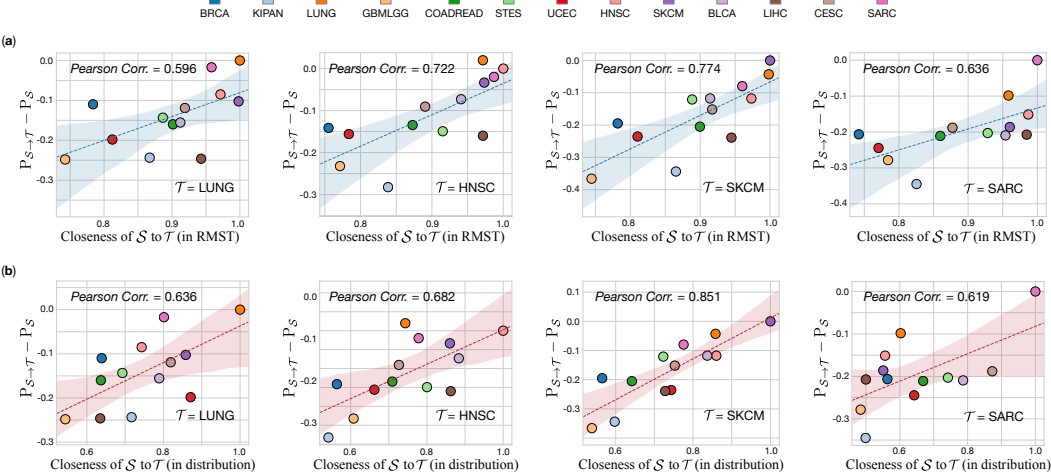

Figure 7: More results on two inter-task factors: (a) $C_{S\to T}^{\text{RMST}}$ and (b) $C_{S\to T}^{\text{Dist.}}$.

**Case Study on Negative Transfer** Taking SARC as an example, most models from other cancers cannot generalize well to it. Beside intra-task factors (*e.g.*, $P_T = 0.531$), one inter-task factor $C_{S\to T}^{\text{Dist.}}$ could also result in this failure. As shown in Figure 7(b), compared to other targets (LUNG, HSNC, and SKCM), SARC exhibits a larger distance from other cancers. Concretely, when $T =$ SARC, there are more points at the places away from $T$, with $C_{S\to T}^{\text{Dist.}} < 0.8$.

**Collinearity Analysis** As shown in Figure 8, we find a moderate correlation (Pearson correlation = 0.52) between $C_{S\to T}^{\text{RMST}}$ and $C_{S\to T}^{\text{Dist.}}$. This indicates a certain degree of collinearity between two inter-task factors. Moreover, a closer invasiveness may result in a more similar data distribution.

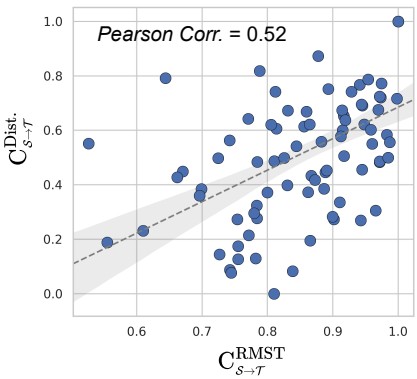

Figure 8: Regression plot between two inter-task factors.

# D ADDITIONAL EXPERIMENTS FOR ROUPKT

## D.1 ABLATION STUDY

ROUPKT mainly consists of a router and an MoE with multiple off-the-shelf, cancer-specific models as prognostic experts (*i.e.*, $E_T$ and $\{E_S \mid S \in C \setminus \{T\}\}$). We conduct ablation studies on these key components and present their results in Table 5.

Concretely, we examine the different combinations of experts in MoE, as well as the router network. **(1)** The first combination is only using the models transferred from other cancers, namely, $E_{\mathcal{T}}$ is excluded. This leads to a decrease of 1.45% in average performance, implying the positive role of $E_{\mathcal{T}}$ in the target task. **(2)** The second combination is only retaining the models with positive performance in transferring to $\mathcal{T}$. It also leads to worse average performance. This could be due to that negative transfer only reflects the *overall performance* of transferring $E_{\mathcal{S}}$ to $\mathcal{T}$; it is possible for $E_{\mathcal{S}}$ to yield good results for some specific samples. The routing strategy could address these specific cases. **(3)** Besides, we adopt MeanMIL (that simply takes the mean of all instance features as the bag representation) to implement the router network of ROUPKT, which obtains an average performance comparable to ABMIL.

Table 5: Ablation study on ROUPKT. $E_{\mathcal{S}}$ = "pos" means that the MoE of ROUPKT only contains the $E_{\mathcal{S}}$ with positive transfer.

| Components | | | Performance on $\mathcal{T}$ | | | | | | | | | | | | | Avg. |
| Router | $E_{\mathcal{S}}$ | $E_{\mathcal{T}}$ | BRCA | KIPAN | LUNG | GBM LGG | COAD READ | STES | UCEC | HNSC | SKCM | BLCA | LIHC | CESC | SARC | |
| --- | --- | --- | --- | --- | --- | --- | --- | --- | --- | --- | --- | --- | --- | --- | --- | --- |
| ABMIL | all | ✓ | 0.7181 | 0.8096 | 0.5714 | 0.7726 | 0.7123 | 0.6708 | 0.7371 | 0.6257 | 0.5954 | 0.6644 | 0.7563 | 0.6629 | 0.5596 | 0.6812 |
| ABMIL | all | | 0.7101 | 0.7894 | 0.5630 | 0.7723 | 0.6911 | 0.6532 | 0.7413 | 0.6141 | 0.5838 | 0.6621 | 0.7365 | 0.6763 | 0.5337 | 0.6713 |
| ABMIL | pos | ✓ | 0.7095 | 0.8079 | 0.5561 | 0.7726 | 0.6981 | 0.6725 | 0.7371 | 0.6257 | 0.5992 | 0.6644 | 0.7266 | 0.6519 | 0.5508 | 0.6748 |
| MeanMIL | all | ✓ | 0.7361 | 0.8171 | 0.5816 | 0.7739 | 0.6970 | 0.6684 | 0.7756 | 0.6428 | 0.5907 | 0.6329 | 0.7370 | 0.6564 | 0.5635 | **0.6826** |

### D.2 HYPER-PARAMETER ANALYSIS

**Different Settings of $K$** The experimental results of ROUPKT with different $K$ (*i.e.*, 3, 5, 7, and 13) are shown in Table 6. Since MoE is often set for sparse activation (Jacobs et al., 1991; Shazeer et al., 2017), we mainly test small values for $K$. A setting of $K = 13$ is specially examined because it represents an extreme case in which no routing is applied and all available experts are aggregated just like the popular attention mechanism (Ilse et al., 2018). However, its result suggests that such way is not as good as expert routing, which again demonstrates the superiority of routing-based strategies in utilizing and combining multiple experts.

Table 6: Performance of ROUPKT with different $K$ (determining the number of experts to use).

| Settings | Performance on $\mathcal{T}$ | | | | | | | | | | | | | Avg. |
| | BRCA | KIPAN | LUNG | GBM LGG | COAD READ | STES | UCEC | HNSC | SKCM | BLCA | LIHC | CESC | SARC | |
| --- | --- | --- | --- | --- | --- | --- | --- | --- | --- | --- | --- | --- | --- | --- |
| $K = 3$ | 0.6980 | 0.8002 | 0.5647 | 0.7726 | 0.6997 | 0.6699 | 0.7389 | 0.6253 | 0.5918 | 0.6626 | 0.7389 | 0.6823 | 0.5461 | 0.6762 |
| $K = 5$ | 0.7181 | 0.8096 | 0.5714 | 0.7726 | 0.7123 | 0.6708 | 0.7371 | 0.6257 | 0.5954 | 0.6644 | 0.7563 | 0.6629 | 0.5596 | **0.6812** |
| $K = 7$ | 0.7085 | 0.8109 | 0.5738 | 0.7734 | 0.7030 | 0.6703 | 0.7429 | 0.6211 | 0.5807 | 0.6601 | 0.7467 | 0.6647 | 0.5347 | 0.6762 |
| $K = 13$ | 0.7087 | 0.8073 | 0.5538 | 0.7740 | 0.6849 | 0.6720 | 0.7508 | 0.6235 | 0.5865 | 0.6402 | 0.7271 | 0.6520 | 0.5422 | 0.6710 |

**Different Coefficients of $\mathcal{L}_Z$** $\mathcal{L}_Z$ is used to penalize the extremely-large scores in expert routing. Therefore, it could encourage ROUPKT to leverage the WSI-based prognostic knowledge from diverse off-the-shelf experts, rather than only the best one. The results of different coefficients are exhibited in Table 7. We observe that a larger coefficient tends to result in a better overall performance. This result indicates that ROUPKT could often perform better when it is able to benefit from diverse prognostic knowledge.

Table 7: Performance of ROUPKT with different coefficients (coef.) of $\mathcal{L}_Z$.

| Coef. of $\mathcal{L}_Z$ | Performance on $\mathcal{T}$ | | | | | | | | | | | | | Avg. |
| | BRCA | KIPAN | LUNG | GBM LGG | COAD READ | STES | UCEC | HNSC | SKCM | BLCA | LIHC | CESC | SARC | |
| --- | --- | --- | --- | --- | --- | --- | --- | --- | --- | --- | --- | --- | --- | --- |
| 0 | 0.7473 | 0.8029 | 0.5679 | 0.7730 | 0.6773 | 0.6637 | 0.6969 | 0.6249 | 0.5980 | 0.6642 | 0.7529 | 0.6622 | 0.5211 | 0.6732 |
| 0.001 | 0.7227 | 0.7995 | 0.5667 | 0.7731 | 0.7081 | 0.6640 | 0.7232 | 0.6238 | 0.5940 | 0.6697 | 0.7555 | 0.6733 | 0.5168 | 0.6762 |
| 0.005 | 0.7186 | 0.8021 | 0.5750 | 0.7729 | 0.7107 | 0.6645 | 0.7233 | 0.6246 | 0.5938 | 0.6670 | 0.7551 | 0.6603 | 0.5390 | 0.6775 |
| 0.01 | 0.7181 | 0.8096 | 0.5714 | 0.7726 | 0.7123 | 0.6708 | 0.7371 | 0.6257 | 0.5954 | 0.6644 | 0.7563 | 0.6629 | 0.5596 | **0.6812** |

### D.3 MORE RESULTS ON EXPERT ROUTING

**Case Study and Visualization** Here, we examine the expert routing in ROUPKT to see how transferred prognostic knowledge is utilized and combined. As shown in Figure 9, we find that expert routing tends to assign higher scores to the experts who can capture more helpful information.

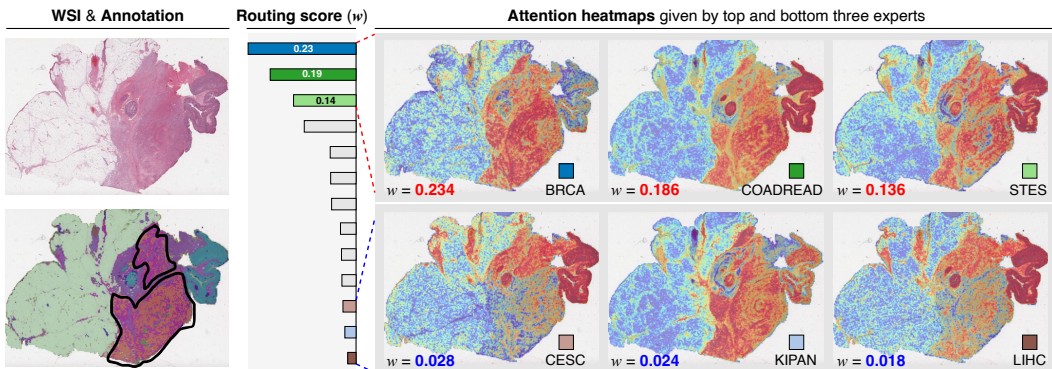

Figure 9: A case study on expert routing. This WSI sample is from the test set of COADREAD.

Besides the above result, we show two additional cases in Figure 10. From these results, we also see that the expert routing in ROUPKT can identify helpful experts (that capture prognosis-relevant tissue regions) and assign higher scores to them. In addition, for different inputs, it produces different combinations of transferred knowledge, rather than a fixed one. These results further show the effectiveness and diversity of expert routing.

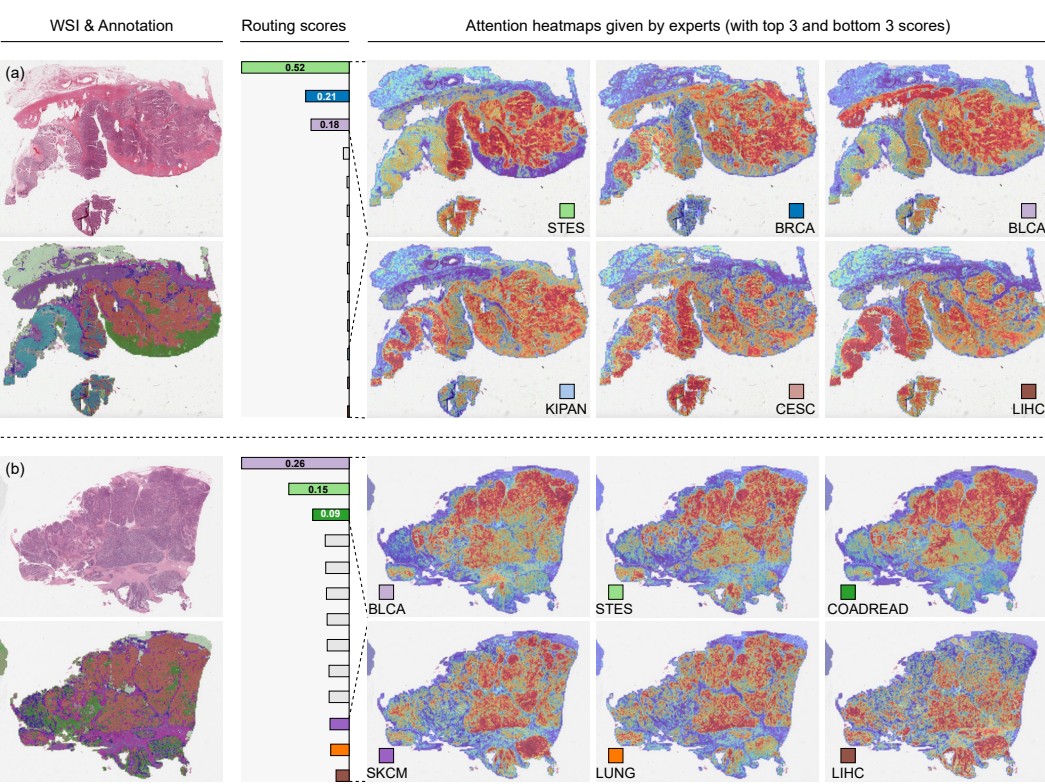

Figure 10: More case studies on expert routing in ROUPKT. The two WSI samples above are from the test set of COADREAD.

**Dynamics of Routing Scores in Training**   Apart from the above case studies on expert routing, we also examine the overall change of routing scores during training, using $\mathcal{T}$ = COADREAD as an example. As shown in Figure 11, we observe that the experts from STES, BRCA, and BLCA often achieve higher average routing scores than the others. These experts also receive a good performance (C-Index = 0.636, 0.604, and 0.597, respectively) in model transfer, as shown in Figure 2. This result implies that the model with better transfer performance is more likely to receive higher

average scores in routing. However, this is not always the case. For example, the model from KIPAN and COADREAD show a C-Index of 0.617 and 0.673, respectively; yet their routing scores are often near zero. One possible reason is that these models could only offer redundant features that those experts with top routing scores have already produced.

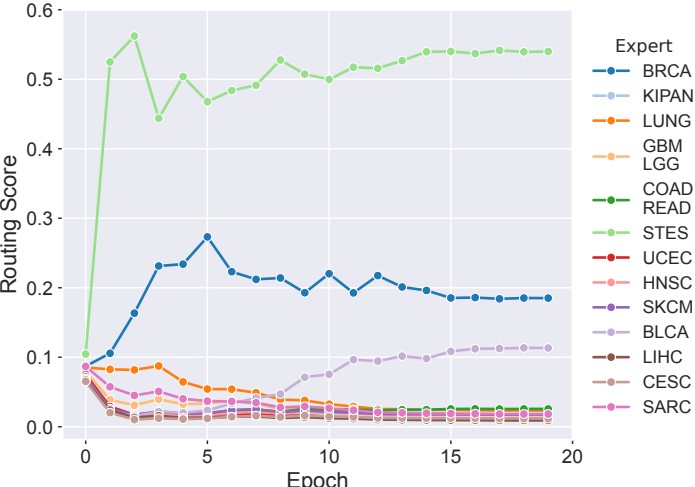

Figure 11: Dynamics of the routing score of all experts in training ($\mathcal{T}$ = COADREAD). This score is averaged over all training samples for each epoch.

## D.4 OTHER MIL MODELS AS EXPERTS

Since any MIL-based models can be easily integrated into the proposed ROUPKT framework, we apply ROUPKT to MIL models other than ABMIL to examine the performance gain attributed to knowledge transfer when varying the architecture of experts. Specifically, we implement two representative MIL models in WSI-based survival analysis, *i.e.*, Patch-GCN (Chen et al., 2021) and TransMIL (Shao et al., 2021). We apply ROUPKT to them by freezing their MIL encoders and employing these encoders as the experts in ROUPKT. All hyper-parameters of ROUPKT remain the same as those used in Table 2.

Table 8: Results of employing other MIL models as the experts in ROUPKT.

| Model | BRCA | KIPAN | LUNG | GBM LGG | COAD READ | STES | UCEC | HNSC | SKCM | BLCA | LIHC | CESC | SARC | Avg. |
|---|---|---|---|---|---|---|---|---|---|---|---|---|---|---|
| \multicolumn{15}{l}{- w/ **Patch-GCN** models (Chen et al., 2021)} |
| $\mathcal{M}_\mathcal{T}$ | 0.7017 (± 0.024) | 0.8181 (± 0.009) | 0.5850 (± 0.027) | 0.7853 (± 0.019) | 0.6675 (± 0.041) | 0.6575 (± 0.035) | 0.7369 (± 0.058) | 0.6019 (± 0.037) | 0.5770 (± 0.040) | 0.6164 (± 0.025) | 0.7487 (± 0.062) | 0.6585 (± 0.065) | 0.5736 (± 0.034) | 0.6714 |
| ROUPKT | 0.7389 (± 0.038) | 0.8083 (± 0.019) | 0.5757 (± 0.030) | 0.7797 (± 0.022) | 0.7076 (± 0.045) | 0.6666 (± 0.059) | 0.7441 (± 0.060) | 0.6402 (± 0.031) | 0.5831 (± 0.015) | 0.6520 (± 0.037) | 0.7528 (± 0.050) | 0.6751 (± 0.064) | 0.5808 (± 0.033) | **0.6850** |
| \multicolumn{15}{l}{- w/ **TransMIL** models (Shao et al., 2021)} |
| $\mathcal{M}_\mathcal{T}$ | 0.6802 (± 0.026) | 0.8121 (± 0.018) | 0.5669 (± 0.021) | 0.7873 (± 0.019) | 0.6651 (± 0.059) | 0.6582 (± 0.056) | 0.7471 (± 0.072) | 0.6315 (± 0.014) | 0.5786 (± 0.039) | 0.6173 (± 0.029) | 0.7385 (± 0.044) | 0.6365 (± 0.068) | 0.5152 (± 0.042) | 0.6642 |
| ROUPKT | 0.7261 (± 0.053) | 0.7921 (± 0.010) | 0.5636 (± 0.019) | 0.7836 (± 0.009) | 0.6993 (± 0.036) | 0.6492 (± 0.062) | 0.7394 (± 0.064) | 0.6389 (± 0.040) | 0.5849 (± 0.018) | 0.6617 (± 0.015) | 0.7521 (± 0.049) | 0.6961 (± 0.081) | 0.5954 (± 0.031) | **0.6832** |

From the comparative results shown in Table 8, we can find that ROUPKT obtains 2.03% and 2.86% performance gains over vanilla Patch-GCN and TransMIL models, respectively. This further confirms that ROUPKT often effectively utilizes knowledge from other cancers to improve prognosis performance, even when applied to the other MIL models.

## D.5 FEW-SHOT PERFORMANCE

We simulate a few-shot setting to examine whether ROUPKT can also leverage knowledge from other cancers to improve prognosis performance when the sample size is very limited ($N < 100$).

Specifically, we implement a few-shot scenario where only a subset of training data ($N = 32$, 64 and 96) can be used in training for $\mathcal{T}$. Given the randomness in subset sampling, we run 5 trials and report the median metric for each experiment. In total, there are 1950 runs on 13 datasets for $\mathcal{M}_{\mathcal{T}}$ and ROUPKT. The result of few-shot performance is shown in Table 9.

Table 9: Performance of $\mathcal{M}_{\mathcal{T}}$ and ROUPKT in few-shot scenarios.

| Model | \multicolumn{13}{c}{Performance on $\mathcal{T}$} | Avg. |
|---|---|---|---|---|---|---|---|---|---|---|---|---|---|---|
| | BRCA | KIPAN | LUNG | GBM LGG | COAD READ | STES | UCEC | HNSC | SKCM | BLCA | LIHC | CESC | SARC | |
| - $N = 32$ | | | | | | | | | | | | | | |
| $\mathcal{M}_{\mathcal{T}}$ | 0.5062 | 0.5163 | 0.5162 | 0.7477 | 0.6191 | 0.5536 | 0.7002 | 0.4652 | 0.4857 | 0.5716 | 0.6348 | 0.5801 | 0.5332 | 0.5715 |
| ROUPKT | 0.5263 | 0.5994 | 0.5143 | 0.7364 | 0.6026 | 0.5388 | 0.6673 | 0.5109 | 0.4980 | 0.5395 | 0.6684 | 0.6474 | 0.4882 | **0.5798** |
| - $N = 64$ | | | | | | | | | | | | | | |
| $\mathcal{M}_{\mathcal{T}}$ | 0.5275 | 0.5639 | 0.5174 | 0.7523 | 0.6261 | 0.5437 | 0.7073 | 0.4691 | 0.5021 | 0.5614 | 0.6542 | 0.6110 | 0.5307 | 0.5821 |
| ROUPKT | 0.5307 | 0.5963 | 0.5123 | 0.7588 | 0.6152 | 0.5501 | 0.7151 | 0.5360 | 0.5107 | 0.5411 | 0.6878 | 0.6434 | 0.5156 | **0.5933** |
| - $N = 96$ | | | | | | | | | | | | | | |
| $\mathcal{M}_{\mathcal{T}}$ | 0.5547 | 0.6255 | 0.5188 | 0.7558 | 0.6274 | 0.5565 | 0.7139 | 0.4694 | 0.5157 | 0.5705 | 0.6937 | 0.6422 | 0.5343 | 0.5983 |
| ROUPKT | 0.5778 | 0.7104 | 0.5040 | 0.7641 | 0.6169 | 0.5731 | 0.7331 | 0.5358 | 0.5463 | 0.5605 | 0.7163 | 0.6691 | 0.5493 | **0.6197** |

Our analysis has two folds. (1) ROUPKT often performs better than the cancer-specific model trained on the same limited data. It obtains an overall improvement of 1.45%, 1.92%, and 3.58% over $\mathcal{M}_{\mathcal{T}}$ when $N = 32$, 64 and 96, respectively. This suggests that knowledge transfer via ROUPKT often improves prognosis performance when available training data is limited. (2) Nevertheless, we also note that on some tasks like LUNG and BLCA, ROUPKT cannot obtain better performance in few-shot learning. One possible reason is that ROUPKT includes a router for expert selection and a simple adapter for each frozen MIL encoder, which may require more data to train when the target task is relatively difficult. Combining with the approach tailored for few-shot scenarios, *e.g.*, meta-learning (Munkhdalai & Yu, 2017; Hospedales et al., 2021), could be one possible solution. We leave it as our future work.

## D.6 COMPARISON WITH FEATHER

We further compare the proposed ROUPKT with a recent slide-level foundation model, Feather (Shao et al., 2025). It is an ABMIL-based encoder pretrained on a pan-cancer morphological classification task (108-way classification). We implement its two variants for comparisons. (1) Feather (frozen) + MLP, where the weight of Feather is frozen in training, and MLP is trained as a prediction head for WSI-based cancer prognosis. (2) Feather (free) + MLP, where both Feather and MLP are involved in gradient descent and network optimization. Both Feather models are initialized with the pretrained weights downloaded from the official website [7]. Their results are exhibited in Table 10.

Table 10: Results of comparison with Feather (Shao et al., 2025). [†] The MIL encoder of Feather is also involved in fine-tuning, not in a frozen state.

| Model | \multicolumn{13}{c}{Performance on $\mathcal{T}$} | Avg. |
|---|---|---|---|---|---|---|---|---|---|---|---|---|---|---|
| | BRCA | KIPAN | LUNG | GBM LGG | COAD READ | STES | UCEC | HNSC | SKCM | BLCA | LIHC | CESC | SARC | |
| Feather + MLP | 0.6419 ($\pm$ 0.046) | 0.7560 ($\pm$ 0.034) | 0.5875 ($\pm$ 0.062) | 0.7700 ($\pm$ 0.024) | 0.6592 ($\pm$ 0.040) | 0.6420 ($\pm$ 0.023) | 0.6639 ($\pm$ 0.081) | 0.5733 ($\pm$ 0.066) | 0.5104 ($\pm$ 0.078) | 0.6232 ($\pm$ 0.043) | 0.6709 ($\pm$ 0.056) | 0.5979 ($\pm$ 0.094) | 0.5911 ($\pm$ 0.033) | 0.6375 |
| Feather + MLP [†] | 0.6581 ($\pm$ 0.060) | 0.8150 ($\pm$ 0.035) | 0.5827 ($\pm$ 0.039) | 0.7732 ($\pm$ 0.017) | 0.6574 ($\pm$ 0.049) | 0.6706 ($\pm$ 0.030) | 0.7316 ($\pm$ 0.053) | 0.6048 ($\pm$ 0.053) | 0.5433 ($\pm$ 0.064) | 0.6443 ($\pm$ 0.031) | 0.7079 ($\pm$ 0.054) | 0.6411 ($\pm$ 0.041) | 0.5719 ($\pm$ 0.040) | 0.6617 |
| ROUPKT | 0.7181 ($\pm$ 0.051) | 0.8096 ($\pm$ 0.007) | 0.5714 ($\pm$ 0.033) | 0.7726 ($\pm$ 0.019) | 0.7123 ($\pm$ 0.047) | 0.6708 ($\pm$ 0.042) | 0.7371 ($\pm$ 0.054) | 0.6257 ($\pm$ 0.034) | 0.5954 ($\pm$ 0.025) | 0.6644 ($\pm$ 0.017) | 0.7563 ($\pm$ 0.048) | 0.6629 ($\pm$ 0.061) | 0.5596 ($\pm$ 0.040) | **0.6812** |

We have two notable findings from the above results. (1) ROUPKT can also obtain better overall performance than these baselines, with an improvement of 2.95% over a fully fine-tuned Feather model. This further confirms the potential of knowledge transfer in WSI-based cancer prognosis tasks. (2) In some challenging tasks, *e.g.*, LUNG and SARC, the fully fine-tuned Feather can perform better than ROUPKT. This suggests the utility of knowledge from other sources. Concretely, besides utilizing knowledge from internal sources (*e.g.*, $\mathcal{M}_{\mathcal{T}}$), leveraging knowledge from external diverse sources (*e.g.*, the pretrained Feather) could further benefit the target model. This could be another interesting direction for knowledge transfer studies.

---

[7] https://huggingface.co/MahmoodLab/abmil.base.uni_v2.pc108-24k

## E    LIMITATIONS

Here we discuss the potential limitations of CROPKT. They span two main aspects as follows.

**Limitation in Pan-Cancer Study Cohorts**  This study and its empirical findings primarily rely on the publicly available data from TCGA. Although TCGA has covered a huge number of samples for cancer research and currently holds a leading position in terms of data scale and the diversity of cancer types, it is still lacking in patient diversity, given the large heterogeneity of tumors. Therefore, constructing more diverse cohorts would strengthen this study. Nevertheless, gathering pan-cancer WSI samples remains a significant challenge, due to the gigapixel size of WSIs and concerns about patient privacy.

**Limitation in Experimental Designs** It is discussed below.

- The WSIs used in this study are processed by UNI2-h (Chen et al., 2024). We choose UNI as it provides reusable large-scale patch features to support this study, and it has demonstrated state-of-the-art performance in many downstream pathology tasks. Despite its remarkable performance and high impact in computational pathology, there could be special artifacts caused by the model architecture or training data. This may limit the generalizability of this study's conclusions. Therefore, using another pathology foundation model, *e.g.*, CONCH (Lu et al., 2024) or MUSK (Xiang et al., 2025), would strengthen the core findings of this study.

- It is the same for the choice of MIL networks. This study mainly adopts ABMIL as the backbone network for transferability evaluation. It would be better to extend to more MIL architectures as shown in Shao et al. (2025). Nevertheless, we would like to note that the above different settings (*i.e.*, different pathology foundation models and MIL networks) will lead to large-scale, intensive model training and evaluation, because there are 26 cancer types and hundreds of model transfer tests for study, and every model evaluation involves a five-fold cross-validation.

- A quantitative comparison between knowledge transfer and MTL. It could help to know the capabilities of knowledge transfer and MTL. However, due to limitations in computational hardware, it is not presented in this study. MTL requires large-scale training on a multi-cancer WSI dataset, posing critical computational challenges. One primary issue is hardware requirements: MTL needs to simultaneously process all datasets (503 GB in total for storage in this study) during iterative training, which imposes significantly higher hardware requirements (*e.g.*, RAM and GPU memory) than single-task learning like ROUPKT.

## F    FUTURE WORKS

This paper switches the focus from cancer-specific training and multi-task learning to knowledge transferring and conducts the first systematic study (CROPKT) on cross-cancer prognosis knowledge transfer in WSIs. Instead of seeking another state-of-the-art approach to improve cross-cancer model transfer, this study aims to answer more fundamental questions in this nascent paradigm, *e.g.*, can and why can WSI-based prognostic knowledge be transferred across different cancers? can cross-cancer knowledge be transferred to further improve prognostic performance? We hope CROPKT could lay the foundation for the research on this new topic. Based on the empirical findings uncovered in this paper, we discuss the following future works.

**(1) New computational methods to reduce the impact of the intrinsic gap between different cancers**. Transfer performance is affected by inter-task factors, as shown in Figure 4. According to this finding, new transfer learning methods could be designed to reduce the impact of the intrinsic gap caused by inter-task factors. Domain adaptation could be a promising solution (Zhuang et al., 2020; Guan & Liu, 2021; Li et al., 2024). However, it is studied extensively for applications on natural images, remaining under-explored for WSIs. WSIs often present extremely large intra-tissue and inter-tissue variations, compared with natural images. Thus, new computational approaches tailored for WSIs are still worth exploring. Besides, integrating current domain adaptation techniques into the proposed baseline approach ROUPKT may further improve model performance.

**(2) Computation-efficient strategies to enable multi-task training involving large-scale multi-cancer WSI datasets**. Unlike the model transfer strategies that only make use of the static knowledge in off-the-shelf models, multi-task learning (MTL) could enable *dynamic knowledge sharing*

between different tasks (Zhang & Yang, 2021). Thus, MTL could be an alternative approach to benefiting from the generalizable knowledge of other cancers. However, as stated before, MTL-like solutions usually have high demands on computational resources and costs, as they need to train a model on ultra-large multi-cancer WSI datasets. In this case, computation-efficient strategies need to be studied to unlock the potential of MTL in WSI-based cancer prognosis.

**(3) Cancer grouping strategies to maximize the benefit of multi-task training**. In view of the negative transfer observed in Figure 2 and the near-zero routing scores produced for most experts in Figure 11, how to determine an optimal group from cancer candidates also needs to be taken into account. This issue, known as task grouping in MTL, has also been widely studied in many well-known applications; their findings suggest the importance of task grouping (Zamir et al., 2018; Standley et al., 2020; Fifty et al., 2021). However, it has not yet been studied in computational pathology. Thus, studying how to group cancer-specific tasks could further enhance the performance of WSI-based cancer prognosis.

