# OpenReview forum: "Cross-Cancer Knowledge Transfer in WSI-based Prognosis Prediction"
_ICLR.cc/2026/Conference — Submitted to ICLR 2026_

### Official Review · Reviewer_FJ4V · 2025-10-21

**Soundness:** 3
**Presentation:** 4
**Contribution:** 3
**Rating:** 6
**Confidence:** 4

**Summary:**

This paper tackles the problem that survival prediction from gigapixel histopathology WSIs typically follows a cancer-specific training paradigm, which does not scale to rare tumors and cannot leverage knowledge across cancers. Specifically, the authors curated a pan-cancer dataset UNI2‑h‑DSS and proposed a routing-based mixture-of-experts that adaptively combines frozen, off‑the‑shelf cancer-specific models to improve target survival prediction.

**Strengths:**

1.	Problem framing is timely and relevant for computational pathology: leveraging cross-cancer information is important for rare tumors and for improving generalization.
2.	Systematic transferability evaluation across many cancer pairs is informative. Observing both positive and negative transfer is clinically and methodologically relevant.
3.	ROUPKT is a pragmatic, compute-efficient architecture that reuses frozen per-cancer encoders and learns a small router + per-expert adapters on the target. Training and inference efficiency considerations are appropriate for WSI-scale workloads.
4.	The paper is generally clearly written and scoped.

**Weaknesses:**

1.	OLS appropriateness: The response variable (C-index) is bounded in [0,1] and fold-averaged. The design likely violates OLS assumptions (normality, homoskedasticity, independence).
2.	Few-shot adaptation baselines: Since rare tumors may have very small labeled sets, compare ROUPKT against simple few-shot fine-tuning or adapters on the target, with and without source experts, to contextualize zero-shot vs few-shot practicality.
3.	Claims of efficiency vs MTL should be supported with actual training time, memory, and FLOPs for ROUPKT vs representative MTL and stacking baselines.

**Questions:**

See Weaknesses.

---

> ### Author Response · Authors · 2025-11-28
> **Response to Reviewer FJ4V (1/n)**
>
> Dear Reviewer `FJ4V`,
>
> Thank you for your invaluable time and effort for this review. We are encouraged by your positive evaluation and your recognition of this work in problem framing, transferability evaluation, and methodology.
>
> Also, we are grateful for your insightful comments and constructive suggestions.
>
> We provide a detailed point-to-point response to your questions and concerns, as follows.

---

> ### Author Response · Authors · 2025-11-28
> **Response to Reviewer FJ4V (2/n)**
>
> **W1**: OLS appropriateness: The response variable (C-index) is bounded in [0,1] and fold-averaged. The design likely violates OLS assumptions (normality, homoskedasticity, independence).
>
> ---
>
> **AW1**: Thank you for raising this important point. As per your comment, we have looked into this issue and found that a bounded and fold-averaged C-Index is indeed not an appropriate response variable for OLS regression. Thanks for your kind reminder. We really appreciate your insightful and professional suggestion.
>
> To address your concerns, we have replaced the OLS regression model with a Beta regression model (Ferrari & Cribari-Neto, 2004) to analyze the transfer performance, because Beta regression is more appropriate for a response variable bound in [0, 1] and with heteroskedasticity. With this statistical model, we obtained new results in multivariate analysis, as follows:
>
> **Table 1** Multivariate analysis with Beta regression. It assesses whether factors can explain the variations in transfer performance. $\phi$ is the precision of Beta distribution. NLL = Negative Log Likelihood. $^\ast$ P-value $\le$ 0.05.
> | $\mathrm{P}_{\mathcal{S}}$ | $\mathrm{P}_{\mathcal{T}}$ | $\mathrm{C}_{\mathcal{S}\to\mathcal{T}}^{\text{RMST}}$ | $\mathrm{C}_{\mathcal{S}\to\mathcal{T}}^{\text{Dist.}}$ | $\phi$ ($\uparrow$) | NLL ($\downarrow$) | P-value |
> |:--:|:--:|:--:|:--:|:--:|:--:|:--:|
> | $\checkmark$ | | | | 3.617 | -186.2 |  |
> | | $\checkmark$ | | | 3.726 | -195.3 | $\ast$ |
> | $\checkmark$ | $\checkmark$ | | | 3.742 | -196.7 | /$\ast$ |
> | $\checkmark$ | $\checkmark$ | $\checkmark$ | | 3.745 | -196.9 | /$\ast$/ |
> | $\checkmark$ | $\checkmark$ | | $\checkmark$ | 3.866 | -207.2 | $\ast$/$\ast$/$\ast$ |
> | $\checkmark$ | $\checkmark$ | $\checkmark$ | $\checkmark$ | **3.883** | **-208.6** | $\ast$/$\ast$/ /$\ast$ |
>
> Our analysis for the above result is as follows:
> - $\mathrm{P}_{\mathcal{S}}$: It is not a significant factor in Beta regression. This reveals that only the performance of source model cannot explain the variation of transfer performance. There are other important factors.
> - $\mathrm{P}_{\mathcal{T}}$: It is a significant one. This indicates that the transfer performance could be affected more by the target task, compared with the source model.
> - $\mathrm{C}_{\mathcal{S}\to\mathcal{T}}^{\text{RMST}}$: The statistical result in Table 1 shows that it helps to explain the variation to some extent, though not significantly.
> - $\mathrm{C}_{\mathcal{S}\to\mathcal{T}}^{\text{Dist.}}$: Multivariate analysis suggests that it is a critical factor affecting knowledge transfer (P-value $<0.05$).
> - Finally, combining all intra-task and inter-task factors yields the best performance in Table 1. These results suggest that the four defined factors help explain cross-cancer transfer performance with varying degrees of contribution.
>
> **Our revisions**: In revision, we have replaced the OLS results with the new result exhibited above. The relevant descriptions have also been revised, as stated above. These revisions are colored in red. Please refer to Table 1 on page 7 and lines 320-345 on pages 6 and 7 for details.
>
> Thank you again for your invaluable comments. We are so lucky to have such an exceptional reviewer like you.
>
> ---
>
> [Reference A] Ferrari & Cribari-Neto, Beta regression for modelling rates and proportions. Journal of applied statistics, 2004.

---

> ### Author Response · Authors · 2025-11-28
> **Response to Reviewer FJ4V (3/n)**
>
> **W2**: Few-shot adaptation baselines: Since rare tumors may have very small labeled sets, compare ROUPKT against simple few-shot fine-tuning or adapters on the target, with and without source experts, to contextualize zero-shot vs few-shot practicality.
>
> ---
>
> **AW2**: Thank you for your insightful comments. Reviewer ` SJmc` also raised this concern. The proposed ROUPKT is a framework for knowledge transfer but is not examined in a low-data scenario where knowledge transfer is expected to be impactful. This is our omission in experimental designs. We appreciate your kind remainder and good suggestion.
>
> According to your suggestion, we have implemented a few-shot scenario where only a subset of data ($N=$ 32, 64 and 96) can be used in training for $\mathcal{T}$. Given the randomness in subset sampling, we run 5 trials and report the median metric for each experiment. In total, there are 1950 runs on 13 datasets for $\mathcal{M}_{\mathcal{T}}$ and ROUPKT. The result of few-shot performance is summarized below.
>
> **Table 9** Performance of $\mathcal{M}_{\mathcal{T}}$ and ROUPKT in few-shot scenarios.
> | Model | BRCA | KIPAN | LUNG | GBMLGG | COADREAD | STES | UCEC | HNSC | SKCM | BLCA | LIHC | CESC | SARC | **Avg.** |
> |:--:|:--:|:--:|:--:|:--:|:--:|:--:|:--:|:--:|:--:|:--:|:--:|:--:|:--:|:--:|
> | - $N=32$ |
> | $\mathcal{M}_{\mathcal{T}}$ | 0.5062 | 0.5163 | 0.5162 | 0.7477 | 0.6191 | 0.5536 | 0.7002 | 0.4652 | 0.4857 | 0.5716 | 0.6348 | 0.5801 | 0.5332 | **0.5715** |
> | ROUPKT | 0.5263 | 0.5994 | 0.5143 | 0.7364 | 0.6026 | 0.5388 | 0.6673 | 0.5109 | 0.4980 | 0.5395 | 0.6684 | 0.6474 | 0.4882 | **0.5798** |
> | - $N=64$ |
> | $\mathcal{M}_{\mathcal{T}}$ | 0.5275 | 0.5639 | 0.5174 | 0.7523 | 0.6261 | 0.5437 | 0.7073 | 0.4691 | 0.5021 | 0.5614 | 0.6542 | 0.6110 | 0.5307 | **0.5821** |
> | ROUPKT | 0.5307 | 0.5963 | 0.5123 | 0.7588 | 0.6152 | 0.5501 | 0.7151 | 0.5360 | 0.5107 | 0.5411 | 0.6878 | 0.6434 | 0.5156 | **0.5933** |
> | - $N=96$ |
> | $\mathcal{M}_{\mathcal{T}}$ | 0.5547 | 0.6255 | 0.5188 | 0.7558 | 0.6274 | 0.5565 | 0.7139 | 0.4694 | 0.5157 | 0.5705 | 0.6937 | 0.6422 | 0.5343 | **0.5983** |
> | ROUPKT | 0.5778 | 0.7104 | 0.5040 | 0.7641 | 0.6169 | 0.5731 | 0.7331 | 0.5358 | 0.5463 | 0.5605 | 0.7163 | 0.6691 | 0.5493 | **0.6197* |
>
> Our analysis for the above results has two folds.
>
> (1) ROUPKT often performs better than the cancer-specific model trained on the same limited data. It obtains an overall improvement of 1.45%, 1.92%, and 3.58% over $\mathcal{M}_{\mathcal{T}}$ when $N=$ 32, 64 and 96, respectively. This suggests that knowledge transfer via ROUPKT often improves prognosis performance when available training data is limited ($N<100$).
>
> (2) Nevertheless, we also notice that on some tasks like LUNG and BLCA, ROUPKT cannot obtain better performance in few-shot learning. One possible reason is that ROUPKT has a router for expert selection and a simple adapter for each frozen expert, which may require more data to train when the target task is relatively difficult. Combining with the approach tailored for few-shot scenarios, *e.g.*, meta-learning (Munkhdalai *et al.*, 2017; Hospedales *et al.*, 2021), could be one possible solution. We’d like to leave it as our future work.
>
> **Our revisions**: As a response to your concerns, we have added a new subsection “Few-Shot Performance” and have provided the experimental results above. For more details, please refer to Appendix D.5 Few-Shot Performance on pages 21 and 22 in our revised paper, the red texts at lines 1131-1159.
>
> ---
>
> [Reference B] Munkhdalai et al., Meta Networks. ICML 2017.
>
> [Reference C] Hospedales et al., Meta-learning in neural networks: A survey. IEEE TPAMI, 2021.

---

> ### Author Response · Authors · 2025-11-28
> **Response to Reviewer FJ4V (4/n)**
>
> **W3**: Claims of efficiency vs MTL should be supported with actual training time, memory, and FLOPs for ROUPKT vs representative MTL and stacking baselines.
>
> ---
>
> **AW3**: Thank you for kindly pointing out this.
>
> We understand your concerns about a quantitative comparison between ROUPKT and MTL baselines in terms of computation efficiency. Performing this would help to strengthen the claims. Unfortunately, due to the limitations in computing hardware, we are unable to provide such a quantitative comparison at this time, though we’d like to do so.
>
> Meanwhile, we also would like to kindly point out that, besides commonly-used evaluation metrics like training time, memory, and FLOPs, there is a more fundamental gap between ROUPKT and MTL baselines in hardware requirements:
> - RAM size: There are 13 datasets used in our study. Their storage size (after processed into patch features) ranges from 10G (CESC) to 82G (GBMLGG), 503G in total. For a smooth training, it is expected to load training data into RAM to reduce the time cost in frequent I/O. Therefore, compared with the ROUPKT that performs training only on a single dataset (82G), the MTL that involves iterative training on multi-cancer datasets (503G) has significantly higher requirements for RAM size.
> - GPU memory: Compared with ROUPTK, MTL usually requires a larger batch size to ensure the task diversity in mini-batch. In a recent MTL study (Yuan *et al.*, 2025) on WSI-base cancer prognosis, a batch size of 256 is used, while it is 16 in ROUPKT. This means that MTL often requires a larger GPU memory than single-task learning for WSI data. Moreover, Yuan *et al.* use an A6000 GPU while our experiments are run on a much cheaper RTX 3090 GPU.
>
> To summarize, there is a significant gap between ROUPKT and MTL baselines in hardware requirements. However, in terms of *accumulated* training time and FLOPs on all involved tasks, there may not be a noticeable gap between them. Certainly, it would be better to have a quantitative comparison head-to-head, as you noted.
>
> **Our revision**: As a response to your concerns, we have made revisions as follows. (1) We have added a table that shows the statistics of 13 datasets, in order to let readers have an overview of dataset scale. Please refer to Table 4 on page 15 for details. (2) We have added a discussion in Appendix E Limitations to discuss the concern on a quantitative comparison between ROUPKT and MTL baselines. It is colored in red and can be found at lines 1213-1219 on page 23 in our revised paper.
>
> [Reference D] Yuan et al., Pancancer outcome prediction via a unified weakly supervised deep learning model. Signal transduction and targeted therapy, 2025.
>
> ---
>
> Thank you again for your positive score and insightful feedback.
>
> Your encouraging evaluations motivate us to improve this work further and step along this way with more confidence. Thank you so much!

---

### Official Review · Reviewer_oPrr · 2025-10-22

**Soundness:** 3
**Presentation:** 2
**Contribution:** 4
**Rating:** 4
**Confidence:** 5

**Summary:**

This work explores the transferability of models across organs and tasks for survival analysis. The authors use the state-of-the-art UNI2-h patch encoder to curate an extensive dataset to thoroughly study this question. They show the mechanism behind this transfer through attention heatmaps, and examining tumor-specific factors that are predictive of a task’s transferability. Lastly, the authors propose a mixture of experts approach to utilize each pretrained model as an expert, demonstrating substantially improved performance from this pan-cancer model. While the breadth of experiments is extensive and the MoE method is interesting, the paper is unfortunately limited by somewhat poor writing quality. I would be willing to increase my score if my concerns are addressed.

**Strengths:**

- Explainability with attention heatmaps and correlation analysis are interesting and insightful.
- Evaluation is extensive, performed across 26 cancer types.
- The routing method for fusing models is novel and interesting. Benchmarking against other ensemble approaches is also extensive.
- The authors will release their UNI-2-h pan-cancer feature dataset, reducing the barrier for additional works on this topic.

**Weaknesses:**

- The writing quality is quite poor, in terms of both grammar and flow. I strongly recommend the authors work closely with an LLM to improve the writing quality while maintaining the desired meaning.
- Line 203: Positive transfer should be compared against random initialization and/or mean pooling. With good feature encoders, even these simple baselines can achieve competitive performance.
- Line 210: The existence of positive transfer for a single task is not necessarily indicative of transferability without accounting for multiple hypotheses. For instance, the likelihood that 10 random classifiers to all have transfer performance <0.5 is 0.5^10
- Line 215: Why is C-index of 0.6 the cutoff?
- More detail is needed on Figure 2 caption. How is classification performed? KNN? Is the classification head from the previous task used?
- A benchmark majority vote of ensembled models should also be included in Table 3.
- While Shao et al 2025 only performed classification evaluation, their published slide foundation model (Feather) should be benchmarked against.

Textual edits:
Line 46: insufficient detail
Line 51: What does “undesirable performance” mean?
Line 54: needs citation
Introduction needs discussion of slide foundation models as a means of model transfer.
Line 66-67: Please define knowldege transfer clearly. What is the “switch in focus?”
The heatmap for Figure 2 should be task-specific (i.e row-wise)
Line 307-308: Describe how distance is measured
Clarify R and C in the Figure 1 caption.

**Questions:**

In Figure 3, I would be interested to see a quantitative report of attention overlap between the transferred model and the target model for a task, and report whether performance correlates with this degree of overlap.

---

> ### Author Response · Authors · 2025-11-28
> **Response to Reviewer oPrr (1/n)**
>
> Dear Reviewer `oPrr`,
>
> Thank you for your invaluable time and effort for this review. We are encouraged by your recognition of this work, such as extensive experiments, interesting method, and excellent contribution.
>
> Also, we genuinely appreciate your careful reading and constructive suggestions that guide us to improve this work further. Hats off!
>
> We provide a detailed point-to-point response to your questions and concerns, as follows.

---

> ### Author Response · Authors · 2025-11-28
> **Response to Reviewer oPrr (2/n)**
>
> **W1**: The writing quality is quite poor, in terms of both grammar and flow. I strongly recommend the authors work closely with an LLM to improve the writing quality while maintaining the desired meaning.
>
> ---
>
> **AW1**: Thanks for your feedback and helpful suggestion.
>
> We tried our best to notice the grammar and flow in writing. However, sometimes our non-native language background may lead to some unidiomatic expressions. We believe that these possible flaws don’t usually case too much difficulty in reading and understanding our paper, as commented by other three reviewers: “Another significant strength is the clarity of the writing and presentation” (`SJmc`), “The paper is quite well-written and easy to follow” (`byot`), and “The paper is generally clearly written” (`FJ4V`). We could easily eliminate these flaws with LLMs, as you suggested. Thank you again for your valuable suggestion.
>
> According to your feedback, we have utilized LLMs to assist us in polishing the writing and correcting the grammar. Moreover, we have read throughout the paper and have carefully checked every sentence to ensure that the meaning is maintained as before. Some examples of writing polish or grammar correction are provided as follows:
> - “To mitigate this, it is highly anticipated that the prognostic knowledge from other cancers could be leveraged to enhance generalizability.” (Its previous version: “*To remedy this, it is anticipated to utilize the generalizable prognostic knowledge of other cancers.*”)
> - “However, this introduces an ultra-large WSI dataset and extensive, large-scale training, which require computational resources orders of magnitude greater than single-task learning.” (Its previous version: “*However, they usually require expensive computational resources and costs since each WSI has a gigapixel size and multi-cancer WSIs incur iterative training with ultra-large data.*”)
> - “Knowledge transfer offers a potential alternative to large-scale training on multi-cancer WSI datasets by leveraging multiple fitted, off-the-shelf cancer-specific models or slide-level foundation models. This approach may emerge as a considerably more cost-efficient and effective strategy for harnessing insights from diverse cancer types.” (Its previous version: “*Knowledge transfer can escape from large-scale training on multi-cancer WSI datasets by employing multiple off-the-shelf, fitted cancer-specific models, potentially developing as an apparently much cheaper and more efficient way to benefit from other cancers.*”)
>
> All textual revisions have been colored throughout the paper. Please refer to our revised paper for more details. Thank you again for your good suggestion.

---

> ### Author Response · Authors · 2025-11-28
> **Response to Reviewer oPrr (3/n)**
>
> **W2**: Line 203: Positive transfer should be compared against random initialization and/or mean pooling. With good feature encoders, even these simple baselines can achieve competitive performance.
>
> ---
>
> **AW2**: Thank you for your thoughtful comment. We agree that random initialization is a reasonable criterion for positive transfer. Intuitively, random initialization represents a model without discriminative ability because of its random, unordered behavior in prediction. As a result, a model that can perform better than random initialization implies a discriminative model and positive transfer.
>
> We’d like to clarify that the criterion used in this study (*i.e.*, C-Index = 0.5) is equivalent to the converged version of random initialization, because a C-Index (Concordance Index) of 0.5 indicates sufficiently-random guess in prediction. In general, random initialization should be performed multiple times to ensure sufficient randomness, as
> - the generated number is inherently a pseudo-random number,
> - and the patient number in the target cohort is usually not so large (< 1,000).
>
> When the number of sampling times is sufficient, the prediction of each patient is totally random. In this case, no certain ranking patterns can be found in patient-level survival predictions. This leads to a C-Index of 0.5 since C-Index measures the model’s ability in ranking patients with higher risk before those with lower risk (Harrell *et al.*, 1982).
>
> Nevertheless, we’d like to discuss more and hold an open mind for the criterion for positive transfer, because we have not yet found any reference or literature that provides a formal and unified standard to define positive transfer. It often depends on the scenario in existing studies (Zhang *et al.*, 2020; Zamir *et al.*, 2018; Shao *et al.*, 2025).
>
> ---
>
> [Reference A] Harrell et al., Evaluating the yield of medical tests. JAMA, 1982.
>
> [Reference B] Zhang et al., A comprehensive survey on transfer learning. Proceedings of the IEEE, 2020.
>
> [Reference C] Zamir et al., Taskonomy: Disentangling task transfer learning. CVPR 2018.
>
> [Reference D] Shao et al., Do Multiple Instance Learning Models Transfer? ICML 2025.

---

> ### Author Response · Authors · 2025-11-28
> **Response to Reviewer oPrr (4/n)**
>
> **W3**: Line 210: The existence of positive transfer for a single task is not necessarily indicative of transferability without accounting for multiple hypotheses. For instance, the likelihood that 10 random classifiers to all have transfer performance <0.5 is 0.5^10
>
> ---
>
> **AW3**: Thank you for your careful reading and kind reminder. We understand your concerns about our conclusion regarding transferability.
>
> At line 210, our intention is to express that there is **a certain degree** of transferability across **a range** of cancer diseases, but not to conclude a complete transferability across all involved cancer types. Because multiple hypotheses must be taken into account, as you kindly noted above.
>
> **Our revisions**: To address your concerns, we have rephrased the conclusion to make it clearer and more rigorous: “This suggests a certain degree of transferability across a range of cancer diseases”. This revision is colored in red. Please refer to lines 215-216 on pages 4 and 5 for details.
>
> Thank you again for kindly pointing out this.

---

> ### Author Response · Authors · 2025-11-28
> **Response to Reviewer oPrr (5/n)**
>
> **W4**: Line 215: Why is C-index of 0.6 the cutoff?
>
> ---
>
> **AW4**: Because C-Index ranges from 0.0 to 1.0, a C-Index of 0.6 indicates a barely qualified model for risk discrimination (*i.e.*, discriminating between high and low risk patients), just like a scoring rate of 60% in the exam.
>
> From our understanding, a transferred model shows application potential in prognostic prediction if it is qualified in performance evaluation, *i.e.*, C-Index >= 0.6. If the cutoff is 0.5, it merely means that the model is slightly better than random guess but not qualified. Therefore, we adopt 0.6 as the cutoff. Its purpose is to see how many transferred models show promise in the application of rare tumors.

---

> ### Author Response · Authors · 2025-11-28
> **Response to Reviewer oPrr (6/n)**
>
> **W5**: More detail is needed on Figure 2 caption. How is classification performed? KNN? Is the classification head from the previous task used?
>
> ---
>
> **AW5**: Thank you for kind remainder.
>
> The task of interest in this study is cancer prognosis, also called survival analysis or time-to-event analysis. It predicts a survival function for each patient, namely, the probability of surviving a specified time point $t$. It differs from classical classification problems. Its performance is usually measured by C-Index, which assesses whether a model can predict a higher risk score (derived from survival function) for a patient who dies earlier.
>
> For Figure 2, we perform survival prediction by directly transferring a source model trained on $\mathcal{S}$ to a target cancer $\mathcal{T}$. In performance evaluation, $\mathcal{M}_{\mathcal{S}}$ is evaluated on the five-fold test data from $\mathcal{T}$ to report transfer performance (C-Index), as stated in the second paragraph of Section 4 of our original manuscript.
>
> **Our revisions**: According to your suggestion, we have added more details to the caption of Figure 2: “Figure 2: Performance of cross-cancer knowledge transfer ($\mathcal{S}\to\mathcal{T}$). Cancer-specific models are first trained on their respective datasets and are then transferred to other cancers for survival prediction. $\mathcal{M}_{\mathcal{S}}$ is evaluated on the five-fold test data from $\mathcal{T}$ to report transfer performance (C-Index).”. Please refer to lines 240-242 on page 5, the texts colored in red.
>
> This may help readers quickly grasp the general details of transfer experiments. Thank you again for your constructive suggestion.

---

> ### Author Response · Authors · 2025-11-28
> **Response to Reviewer oPrr (7/n)**
>
> **W6**: A benchmark majority vote of ensembled models should also be included in Table 3.
>
> ---
>
> **AW6**: Thank you for your good suggestion.
>
> Following your comment, we have implemented a majority voting-based model ensemble, as a new baseline added to Table 3. Since our models are not utilized for classification but for survival prediction, the idea of majority voting cannot be directly applied to their ensemble. Thus, we instead implement a soft voting strategy for model ensemble, *i.e.*, averaging over the survival function prediction of all models. Its result is given below:
>
> **Table 3**: Comparison with the other strategies that can utilize and combine multiple experts.
> | Model | BRCA | KIPAN | LUNG | GBMLGG | COADREAD | STES | UCEC | HNSC | SKCM | BLCA | LIHC | CESC | SARC | **Avg.** |
> |:--:|:--:|:--:|:--:|:--:|:--:|:--:|:--:|:--:|:--:|:--:|:--:|:--:|:--:|:--:|
> | Ensemble | 0.5972 | 0.7804 | 0.5322 | 0.7753 | 0.6287 | 0.6146 | 0.7160 | 0.5631 | 0.4976 | 0.6191 | 0.6380 | 0.5987 | 0.4386 | **0.6153** |
> | Routing (in ROUPKT) | 0.7181 | 0.8096 | 0.5714 | 0.7726 | 0.7123 | 0.6708 | 0.7371 | 0.6257 | 0.5954 | 0.6644 | 0.7563 | 0.6629 | 0.5596 | **0.6812** |
>
> From the above comparative results, we can also observe that expert routing is a better approach for learning transferable knowledge from multiple off-the-shelf experts.
>
> **Our revisions**: As a response to your suggestion, we have added the above result to Table 3 and have revised the description of compared baselines. All revisions are colored in red. Please refer to Table 3 on page 9 and lines 429-439 on page 8 for more details.

---

> ### Author Response · Authors · 2025-11-28
> **Response to Reviewer oPrr (8/n)**
>
> **W7**: While Shao et al 2025 only performed classification evaluation, their published slide foundation model (Feather) should be benchmarked against.
>
> ---
>
> **AW7**: Yes, Shao *et al.* (2025) recently presented a surprisingly good slide-level foundation model for pathology. Although it is only evaluated on WSI classification tasks, it could also be adapted to survival analysis and cancer prognosis applications. We really appreciate your insightful comments.
>
> Following your suggestion, we further compare the proposed ROUPKT with the pretrained slide foundation model Feather (Shao *et al.*, 2025). Feather is an ABMIL-based encoder pretrained on a pan-cancer morphological classification task (108-way classification). We implement its two variants for comparisons:
> - Feather (frozen) + MLP: The weight of Feather is frozen in training, and MLP is trained as a prediction head for WSI-based survival analysis.
> - Feather (free) + MLP: Both Feather and MLP are involved in SGD and network optimization.
> Both Feather models are initialized with the pretrained weights downloaded from [the official website]( https://huggingface.co/MahmoodLab/abmil.base.uni_v2.pc108-24k). We show their result as follows:
>
> **Table 10**: Results of comparison with Feather (Shao et al., 2025). * The MIL encoder of Feather is also involved in fine-tuning, not in a frozen state.
> | Model | BRCA | KIPAN | LUNG | GBMLGG | COADREAD | STES | UCEC | HNSC | SKCM | BLCA | LIHC | CESC | SARC | **Avg.** |
> |:--:|:--:|:--:|:--:|:--:|:--:|:--:|:--:|:--:|:--:|:--:|:--:|:--:|:--:|:--:|
> | Feather + MLP | 0.6419 | 0.7560 | 0.5875 | 0.7700 | 0.6592 | 0.6420 | 0.6639 | 0.5733 | 0.5104 | 0.6232 | 0.6709 | 0.5979 | 0.5911 | **0.6375** |
> | Feather + MLP * | 0.6581 | 0.8150 | 0.5827 | 0.7732 | 0.6574 | 0.6706 | 0.7316 | 0.6048 | 0.5433 | 0.6443 | 0.7079 | 0.6411 | 0.5719 | **0.6617** |
> | ROUPKT | 0.7181 | 0.8096 | 0.5714 | 0.7726 | 0.7123 | 0.6708 | 0.7371 | 0.6257 | 0.5954 | 0.6644 | 0.7563 | 0.6629 | 0.5596 | **0.6812** |
>
> There are two notable findings from the above results:
> - ROUPKT can also obtain better overall performance than these baselines, with an improvement of 2.95% over a fully fine-tuned Feather model. This further confirms the potential of knowledge transfer in WSI-based cancer prognosis tasks.
> - In some challenging tasks, *e.g.*, LUNG and SARC, the fully fine-tuned Feather can perform better than ROUPKT. This suggests the utility of cross-cancer prognosis knowledge. Concretely, besides utilizing knowledge from internal sources (*e.g.*, $\mathcal{M}_{\mathcal{T}}$), leveraging knowledge from external diverse sources (*e.g.*, the pretrained Feather) could further benefit the target model. This could be another interesting direction for knowledge transfer studies.
>
> **Our revisions**: According to your comment, we have added the result of Feather models and a detailed analysis for this result. For more details, please refer to Appendix D.6 Comparison with Feather on page 22, the texts colored in red at lines 1160-1187.
>
> We’d like to thank you again for your insightful suggestion. It does drive us to think more about knowledge transfer and motivate us to investigate deeper.

---

> ### Author Response · Authors · 2025-11-28
> **Response to Reviewer oPrr (9/n)**
>
> **W8**: Textual edits: Line 46: insufficient detail Line 51: What does “undesirable performance” mean? Line 54: needs citation Introduction needs discussion of slide foundation models as a means of model transfer. Line 66-67: Please define knowledge transfer clearly. What is the “switch in focus?” The heatmap for Figure 2 should be task-specific (i.e row-wise) Line 307-308: Describe how distance is measured Clarify R and C in the Figure 1 caption.
>
> ---
>
> **AW8**: We are grateful for your careful reading and thorough feedback. Our point-to-point responses are as follows:
>
> (1) To “Line 46: insufficient detail”: This sentence mainly summarizes the issues in current cancer-specific learning paradigm. Its details are explained in the following two points (texts in bold). To make it clearer, we have revised it to “Nevertheless, this paradigm faces inherent limitations that hinder its ability to meet key practical requirements, as follows”. Please refer to lines 46-47 on page 1.
>
> (2) To “Line 51: What does undesirable performance mean?”: Our intention is to express a model with unsatisfactory performance due to limited or low-quality training data. To avoid confusion, we have revised it to “As a result, cancer-specific approaches often yield unsatisfactory models for rare tumor diseases”. Please refer to lines 50-51 on page 1.
>
> (3) To “Line 54: needs citation”: According to your suggestion, we have cited a work by Song *et al*. (Morphological prototyping for unsupervised slide representation learning in computational pathology, CVPR 2024). It highlights the issue of data scarcity and model generalization in current computational pathology in its Introduction on page 1. Please refer to line 54 on page 2.
>
> (4) To “Introduction needs discussion of slide foundation models as a means of model transfer”: Yes, slide foundation models could be another potential means of model transfer, as indicated by the aforementioned additional results in Table 10. Thank you for your insightful comment. We have added it to Introduction: “Knowledge transfer offers a potential alternative to large-scale training on multi-cancer WSI datasets by leveraging multiple off-the-shelf, fitted cancer-specific models or slide-level foundation models (Shao *et al.*, 2025; Ding *et al.*, 2025)”. Please refer to lines 70-73 on page 2.
>
> (5) To “Line 66-67: Please define knowledge transfer clearly”: Based on the classical definition of knowledge transfer in machine learning and the scope of this study, we have provided a clear definition of knowledge transfer in our revised paper: “In this study, knowledge transfer refers to a process where a model trained on one cancer type is transferred and leveraged for a task involving another cancer type.”. Please refer to line 107 on page 2.
>
> (6) To “What is the switch in focus?”: The meaning of this sentence is “shift the focus from cancer-specific or multi-cancer training to knowledge transferring”. To avoid confusion and make it clearer, we have revised it to “Given these limitations, this paper proposes a paradigm shift from cancer-specific or multi-cancer learning to knowledge transferring.”. Please refer to lines 67-68 on page 2.
>
> (7) To “The heatmap for Figure 2 should be task-specific (i.e row-wise)”: Thanks for your suggestion. It seems helpful for better view. We appreciate your careful reading. As a response, we have transposed the heatmaps according to your suggestion. Please refer to the revised Figure 2 on page 5.
>
> (8) To “Line 307-308: Describe how distance is measured”: We have added a description for it: “The distance from $\mathcal{S}$ to $\mathcal{T}$ in data distribution is measured by the Euclidean distance between the centroid of mean-based WSI embeddings in $\mathcal{S}$ and $\mathcal{T}$”. Please refer to lines 318-319 on page 6.
>
> (9) To “Clarify R and C in the Figure 1 caption”: We have added a clarification to the Figure 1 caption: “C and R refer to common and rare cancer diseases”. Please refer to Figure 1 and line 161 on page 3.
>
> All the revisions above are colored in red in our revised paper. We’d like to express a great debt of gratitude to you for your thorough feedback. It does improve the paper substantially. Thank you so much!

---

> ### Author Response · Authors · 2025-11-28
> **Response to Reviewer oPrr (10/n)**
>
> **Q1**: In Figure 3, I would be interested to see a quantitative report of attention overlap between the transferred model and the target model for a task, and report whether performance correlates with this degree of overlap.
>
> **AQ1**: Thanks for your question. That sounds interesting. We have carefully thought about that. Next, we firstly answer it by logical analysis.
>
> From our understanding, for a transferred model $\mathcal{M}_{\mathcal{S}}$, its performance is likely to weakly correlate with the degree of attention overlap between the transferred model and the target model, but tends to highly correlate with the degree of attention overlap between the transferred modeland an ideal target model (*i.e.*, C-Index=1.0). The reason for that is as follows.
>
> We know that, a perfect attention overlap between $\mathcal{M}_{\mathcal{S}}$ and the target model, implies the same transfer performance as that of the target model. But when a difference occurs in the attention maps, the transfer performance cannot be predicted because we don’t know if this difference is helpful for prediction. Only if we know that, we can know the transfer performance has improved or not.
>
> That means, the transfer performance should highly correlate with the degree of attention overlap between $\mathcal{M}_{\mathcal{S}}$ and an ideal target model, because an ideal target model provides perfect attention maps.
>
> Even though, we are also pleased to provide empirical results for the attention overlap analysis if needed. Thank you for your question. It does drive us to think more deeply about the behavior of transferred models.
>
> ---
>
> We feel so fortunate to receive such a great review from you. Your comments are really constructive and insightful, as stated in our response. Thank you again!

---

### Official Review · Reviewer_byot · 2025-10-30

**Soundness:** 2
**Presentation:** 3
**Contribution:** 2
**Rating:** 4
**Confidence:** 4

**Summary:**

The paper studies cross-cancer knowledge transfer for WSI-based prognosis. To mitigate limited sample sizes, it curates a benchmark, analyzes what drives successful transfer across cancer types, and proposes an MoE-based method to exploit transfer. Results support the main claim but would be more convincing with stronger baselines, richer metrics, and deeper analysis.

**Strengths:**

1.	The paper is quite well-written and easy to follow.

2.	The paper proposes using knowledge transfer methods to address the challenge of rare tumors, which is a natural and direct solution.

3.	The experimental results demonstrate the main points of this paper (transfer learning does helps when samples are limited).

**Weaknesses:**

1.	Using only the ABMIL model is questionable. Survival prediction is quite hard; even with gene information, multimodal survival prediction performance is still not fully satisfactory, unlike WSI classification (C-index around 0.5-0.7, few 0.8). The results in Figure 2 are obtained with only the simplest WSI model (ABMIL), which makes it less convincing. To prove the transferability across organs, it would be better if the authors could demonstrate multiple models all converge to similar patterns (not have to be very complicated, TransMIL and MambaMIL would be enough, since they cover major MIL families: Transformer, Mamba, and ABMIL)

2.	The criterion for the experiments in Figure 2 is not fully convincing. There are four available criteria for this: 1) a fully trained model on target, 2) a randomly initialized, untrained model on target, 3) random scores on the target generated by a chosen strategy (for example uniform or Gaussian), and 4) 0.5. The authors chose the fourth choice, but in my view, 2) and 3) would be more rigorous.

3.	Building on the previous point, the authors have not trained a target model initialized with the source weights, and if they do so, it would be fairer to compare those results with the first choice to confirm positive or negative transfer.

4.	The authors should provide more intuitive explanations for the first row of Figure 3, rather than simply saying ‘they are similar’. In addition, there are no such conclusions or guidelines on which cancer types could benefit from transfer (though from the experiments, most of them benefit from transfer).

5.	In Table 2, I suggest reporting baselines in both frozen and fine-tuned settings. ROUPKT freezes experts by design, but the comparison methods should be shown in both modes.

6.	The number of baseline methods is limited. There are plenty of recent works for MIL, and the authors should consider adding some of them to Table 3 for more convincing comparison.

**Questions:**

Besides the questions in weaknesses, I have one additional question for invasiveness.

1.	Why choose invasiveness, why can RMST fully represent invasiveness, and are there references that support this?

---

> ### Author Response · Authors · 2025-11-28
> **Response to Reviewer byot (1/n)**
>
> Dear Reviewer `byot`,
>
> Thank you for your invaluable time and effort for this review. We are encouraged by your recognition of this work’s representation and solution.
>
> Also, we are grateful for your thorough feedback and constructive suggestions.
>
> We provide a detailed point-to-point response to your questions and concerns, as follows.

---

> ### Author Response · Authors · 2025-11-28
> **Response to Reviewer byot (2/n)**
>
> **W1**: Using only the ABMIL model is questionable. Survival prediction is quite hard; even with gene information, multimodal survival prediction performance is still not fully satisfactory, unlike WSI classification (C-index around 0.5-0.7, few 0.8). The results in Figure 2 are obtained with only the simplest WSI model (ABMIL), which makes it less convincing. To prove the transferability across organs, it would be better if the authors could demonstrate multiple models all converge to similar patterns (not have to be very complicated, TransMIL and MambaMIL would be enough, since they cover major MIL families: Transformer, Mamba, and ABMIL)
>
> ---
>
> **AW1**: Thanks for your detailed feedback and good suggestions.
>
> To address your concerns, we have implemented two different MIL models besides ABMIL, *i.e.*, Patch-GCN (a GCN-based model; Chen *et al.*, 2021) and TransMIL (a Transformer-based model; Shao *et al.*, 2021). We choose them because they are frequently adopted and often cast as the most representative methods in WSI-based survival analysis. MambaMIL (Yang *et al.*, 2024) is also an excellent work. However, due to limited time in the rebuttal, we will look into it in the future. Currently, we’d like to prioritize your critical concerns.
>
> Furthermore, we integrate Patch-GCN and TransMIL models into the proposed ROUPKT framework for knowledge transfer, respectively, by freezing their MIL encoders and employing these encoders as the experts in ROUPKT. We train the ROUPKT with the same hyper-parameters as that with ABMIL.
>
> Comparative results are given as follows:
>
> **Table 8**: Results of employing other MIL models as the experts in ROUPKT.
> | Model | BRCA | KIPAN | LUNG | GBMLGG | COADREAD | STES | UCEC | HNSC | SKCM | BLCA | LIHC | CESC | SARC | **Avg.** |
> |:--:|:--:|:--:|:--:|:--:|:--:|:--:|:--:|:--:|:--:|:--:|:--:|:--:|:--:|:--:|
> | - w/ **Patch-GCN** models |
> | $\mathcal{M}_{\mathcal{T}}$ | 0.7017 | 0.8181 | 0.5850 | 0.7853 | 0.6675 | 0.6575 | 0.7369 | 0.6019 | 0.5770 | 0.6164 | 0.7487 | 0.6585 | 0.5736 | **0.6714** |
> | ROUPKT | 0.7389 | 0.8083 | 0.5757 | 0.7797 | 0.7076 | 0.6666 | 0.7441 | 0.6402 | 0.5831 | 0.6520 | 0.7528 | 0.6751 | 0.5808 | **0.6850** |
> | - w/ **TransMIL** models |
> | $\mathcal{M}_{\mathcal{T}}$ | 0.6802 | 0.8121 | 0.5669 | 0.7873 | 0.6651 | 0.6582 | 0.7471 | 0.6315 | 0.5786 | 0.6173 | 0.7385 | 0.6365 | 0.5152 | **0.6642** |
> | ROUPKT | 0.7261 | 0.7921 | 0.5636 | 0.7836 | 0.6993 | 0.6492 | 0.7394 | 0.6389 | 0.5849 | 0.6617 | 0.7521 | 0.6961 | 0.5954 | **0.6832** |
>
> From the above results, we can find that the proposed ROUPKT framework obtains 2.03% and 2.86% performance gains over vanilla Patch-GCN and TransMIL models, respectively. This further confirms that ROUPKT often effectively utilizes knowledge from other cancers to improve prognosis performance, even when applied to the other MIL models. In other words, true benefits of knowledge transfer are observed on three representative models, *i.e.*, ABMIL (in Table 2), Patch-GCN and TransMIL (in Table 8).
>
> **Our revisions**: As a response to your concerns, we have added a new subsection to show the experiments of employing other MIL models as the experts in ROUPKT. The above results in Table 8 are exhibited in this new subsection. For more details, please refer to Appendix D.4 Other MIL Models as Experts, and the texts colored in red at lines 1107-1129 on page 21 in our revised paper.
>
> Thank your again for your good suggestion.
>
> ---
>
> [Reference A] Chen et al., Whole slide images are 2d point clouds: Context-aware survival prediction using patch-based graph convolutional networks. MICCAI 2021.
>
> [Reference B] Shao et al., TransMIL: Transformer based Correlated Multiple Instance Learning for Whole Slide Image Classification. NeurIPS 2021.
>
> [Reference C] Yang et al., MambaMIL: Enhancing Long Sequence Modeling with Sequence Reordering in Computational Pathology. MICCAI 2024.

---

> ### Author Response · Authors · 2025-11-28
> **Response to Reviewer byot (3/n)**
>
> **W2**: The criterion for the experiments in Figure 2 is not fully convincing. There are four available criteria for this: 1) a fully trained model on target, 2) a randomly initialized, untrained model on target, 3) random scores on the target generated by a chosen strategy (for example uniform or Gaussian), and 4) 0.5. The authors chose the fourth choice, but in my view, 2) and 3) would be more rigorous.
>
> ---
>
> **AW2**: Thank you for your thoughtful comment. We agree that 2) and 3) are rigorous criteria. Intuitively, 2) and 3) represent a model with no discriminative ability because of the random, unordered behavior in them. As a result, a model that can perform better than 2) or 3) implies a discriminative model and positive transfer.
>
> We’d like to clarify that the fourth choice (0.5) is equivalent to the converged version of 2) and 3), because a C-Index (Concordance Index) of 0.5 indicates sufficiently-random guess on the target. Taking 3) as an example, Gaussian random scores on the target, it should be performed multiple times to ensure sufficient randomness, as
> - the generated number is inherently a pseudo-random number,
> - and the patient size of the target is usually not so large (< 1,000).
>
> When the number of times is sufficient, the prediction of each patient is totally random. In this case, no certain ranking patterns can be found in patient-level survival predictions. This leads to a C-Index of 0.5 since C-Index measures the model’s ability in ranking patients with higher risk before those with lower risk (Harrell *et al.*, 1982).
>
> In spite of that, we’d like to discuss more and hold an open mind for the criterion because we have not yet found any reference or literature that provides a formal and unified criterion to define positive transfer. It often depends on the scenario in existing studies (Zhang *et al.*, 2020; Zamir *et al.*, 2018; Shao *et al.*, 2025).
>
> We appreciate your insightful feedback.
>
> ---
>
> [Reference D] Harrell et al., Evaluating the yield of medical tests. JAMA, 1982.
>
> [Reference E] Zhang et al., A comprehensive survey on transfer learning. Proceedings of the IEEE, 2020.
>
> [Reference F] Zamir et al., Taskonomy: Disentangling task transfer learning. CVPR 2018.
>
> [Reference G] Shao et al., Do Multiple Instance Learning Models Transfer? ICML 2025.

---

> ### Author Response · Authors · 2025-11-28
> **Response to Reviewer byot (4/n)**
>
> **W3**: Building on the previous point, the authors have not trained a target model initialized with the source weights, and if they do so, it would be fairer to compare those results with the first choice to confirm positive or negative transfer.
>
> ---
>
> **AW3**: Thank you for your suggestions.
>
> In fact, we have tried to train a target model with the source weights in our earlier exploratory experiments, as you mentioned above. However, it turned out that catastrophic forgetting often happens in the target model, leading to
> - a model with the same performance as that initialized with random weights,
> - or a model with serious overfitting on the limited target data.
>
> Thus, we ignore these results and seek another suitable approach for WSI-based knowledge transfer.
>
> We believe that training a target model initialized with the source weights is a reasonable approach for knowledge transfer. However, it seems not applicable to the WSI scenario. One possible reason is that the training data is often very limited for WSIs and most MIL models are relatively small in size. These tend to cause model overfitting or catastrophic forgetting in network training, thereby leading to failure in knowledge transfer evaluation.

---

> ### Author Response · Authors · 2025-11-28
> **Response to Reviewer byot (5/n)**
>
> **W4**: The authors should provide more intuitive explanations for the first row of Figure 3, rather than simply saying ‘they are similar’. In addition, there are no such conclusions or guidelines on which cancer types could benefit from transfer (though from the experiments, most of them benefit from transfer).
>
> ---
>
> **AW4**: Good suggestion. Thank you for kindly pointing out this. More intuitive explanations or analysis could help readers gain a deeper understanding of the transferability.
>
> Generally speaking, similar regions useful for prognosis estimation (such as tumor and stroma tissues) can also be identified by a transferred model, because there are general cellular prognostic patterns across different tissues, as highlighted in Yu *et al.* (2016), Wulczyn *et al.* (2020), and Chen *et al.* (2022). The common patterns include large and irregularly shaped nuclei, unclear boundaries between tumor and normal tissue, *etc*. A well-trained model is able to capture these general prognostic patterns, enabling itself to identify those prognosis-relevant regions in other cancers.
>
> Yes, as you mentioned, there are no such conclusions or guidelines on which cancer types could benefit from transfer. The reasons span two aspects:
> - A clinical oncology aspect: Most oncology studies focus on a specific cancer. They are dedicated to reveal new prognostic biomarkers for a single cancer type. As a result, rare studies pay attention to the transferability between different cancers in the field. However, as people's understanding of different cancer diseases deepens, it becomes well-known that there are general histological biomarkers for various cancers, such as tumor size, tumor infiltration, and cell mitosis. This lays a primary foundation for exploring cross-cancer knowledge transferability in a pure computational way.
> - A computational aspect: Current computational approaches generally follow a cancer-specific paradigm for developing prognostic models, as stated in our paper. Most of them aim to design better representation methods for gigapixel WSIs. Rare studies investigate cross-cancer knowledge transfer in computational pathology. This work presents the first preliminary yet systematic study on this nascent topic. We hope it could inspire more works to explore WSI-based cancer prognosis from a new perspective, *i.e.*, cross-cancer knowledge transfer.
>
> **Our revisions**: To resolve your concerns, we have added a detailed explanation for the first row of Figure 3. It briefly describes the reason behind the identified similar and useful regions. For more details, please refer to the red texts at lines 270-274 on page 6.
>
> Thank you again for your kinder remainder.
>
> ---
>
> [Reference H] Yu et al., Predicting non-small cell lung cancer prognosis by fully automated microscopic pathology image features. Nature Communications, 2016.
>
> [Reference I] Wulczyn et al., Deep learning-based survival prediction for multiple cancer types using histopathology images. PloS one, 2020.
>
> [Reference J] Chen et al., Pan-cancer integrative histology-genomic analysis via multimodal deep learning. Cancer Cell, 2022.

---

> ### Author Response · Authors · 2025-11-28
> **Response to Reviewer byot (6/n)**
>
> **W5**: In Table 2, I suggest reporting baselines in both frozen and fine-tuned settings. ROUPKT freezes experts by design, but the comparison methods should be shown in both modes.
>
> ---
>
> **AW5**: This suggestion sounds great. We really appreciate your comment.
>
> According to your suggestion, we have implemented two additional baselines: $E_{\mathcal{T}}$ + MLP and $E_{\mathcal{S}\to\mathcal{T}}$ + MLP, where $E$ is not frozen and is involved in network fine-tuning. We show their results as follows:
>
> **Table 2**: Comparison with $\mathcal{M}_{\mathcal{T}}$ and other fine-tuned or transferred models. $^*$ The encoder $E$ of these baselines is also involved in fine-tuning, not in a frozen state.
> | Model | BRCA | KIPAN | LUNG | GBMLGG | COADREAD | STES | UCEC | HNSC | SKCM | BLCA | LIHC | CESC | SARC | **Avg.** |
> |:--:|:--:|:--:|:--:|:--:|:--:|:--:|:--:|:--:|:--:|:--:|:--:|:--:|:--:|:--:|
> | $E_{\mathcal{T}}$ + MLP | 0.6723 | 0.8080 | 0.5463 | 0.7744 | 0.6753 | 0.6694 | 0.7136 | 0.6207 | 0.5682 | 0.6384 | 0.7253 | 0.6367 | 0.5462 | **0.6611** |
> | $E_{\mathcal{T}}$ + MLP $^*$ | 0.6304 | 0.7972 | 0.5757 | 0.7657 | 0.6663 | 0.6761 | 0.6880 | 0.6214 | 0.5857 | 0.6211 | 0.7209 | 0.6440 | 0.5034 | **0.6535** |
> | $E_{\mathcal{S}\to\mathcal{T}}$ + MLP | 0.5288 | 0.6430 | 0.5406 | 0.7713 | 0.6345 | 0.6155 | 0.6658 | 0.5759 | 0.5206 | 0.5971 | 0.6023 | 0.6177 | 0.4764 | **0.5992** |
> | $E_{\mathcal{S}\to\mathcal{T}}$ + MLP $^*$ | 0.6309 | 0.8105 | 0.5872 | 0.7771 | 0.6705 | 0.6617 | 0.6843 | 0.6198 | 0.5723 | 0.6356 | 0.7061 | 0.6785 | 0.4936 | **0.6560** |
> | ROUPKT | 0.7181 | 0.8096 | 0.5714 | 0.7726 | 0.7123 | 0.6708 | 0.7371 | 0.6257 | 0.5954 | 0.6644 | 0.7563 | 0.6629 | 0.5596 | **0.6812** |
>
> The above result shows that a fully fine-tuned version of $E_{\mathcal{T}}$ + MLP obtains a degraded performance compared to its partially fine-tuned counterpart. It is due to the overfitting on training samples since the initial weight of $E_{\mathcal{T}}$ is from the well-fitted $\mathcal{M}_{\mathcal{T}}$ and it is trained again on the same samples in fine-tuning.
>
> Moreover, a fully fine-tuned $E_{\mathcal{S}\to\mathcal{T}}$ + MLP obtains substantial gains as its MIL encoder is involved in fine-tuning. However, it performs worse than ROUPKT in terms of overall performance.
>
> **Our revisions**: According to your suggestion, we have added the above new baselines to Table 2 and revised the relevant description in the main paper. For more details, please refer to Table 2 and the texts colored in red at lines 417-418 on page 8.

---

> ### Author Response · Authors · 2025-11-28
> **Response to Reviewer byot (7/n)**
>
> **W6**: The number of baseline methods is limited. There are plenty of recent works for MIL, and the authors should consider adding some of them to Table 3 for more convincing comparison.
>
> ---
>
> **AW6**: Thank you for your comment.
>
> We’d like to clarify that Table 3 aims to compare the strategies that can leverage multiple off-the-shelf experts for knowledge transfer. To be more specific, our purpose is to examine **diverse mechanisms** that can make better use of 13 frozen MIL encoders from different sources for knowledge transfer. Considering this, we compare various representative mechanisms like the popular attention in MIL, recurrent network, self-attention, and expert routing.
>
> Yes, as you mentioned, recent MIL methods could also be adapted and compared. However, the primary purpose of these MIL methods is to aggregate over 1,000 or 10,000 instances into a better bag-level presentation. This fundamentally differs from our original intention, as clarified above. Concretely,
> - Table 3 does not deal with over 1,000 or 10,000 instances, but leverages only 13 frozen MIL encoders;
> - It is oriented towards knowledge transfer, not a classical MIL scenario.
>
> Thank you for giving us the opportunity to elaborate on this.

---

> ### Author Response · Authors · 2025-11-28
> **Response to Reviewer byot (8/n)**
>
> **Q1**: Why choose invasiveness, why can RMST fully represent invasiveness, and are there references that support this?
>
> **AQ1**: We choose invasiveness because it is believed to one of the main factors that affect prognosis estimation (Liu et al., 2018). For a cancer with stronger invasiveness, *e.g.*, brain tumor, the patients often survive less than 3 years. But for a cancer with weaker invasiveness, *e.g.*, breast cancer, most patients could survive more than 10 years. This means that there is an intrinsic, large gap between different cancer types in survival estimating. Accordingly, invasiveness should be taken into account when transferring a prognostic model from one cancer type to another.
>
> By definition, RMST is the average survival time during a defined time period, as a metric commonly used in survival analysis (Kim *et al.*, 2017). When patients have a cancer disease with stronger invasiveness, their overall survival time would be shorter. This can be reflected by RMST. As shown in Figure 4 (c) right, breast cancer has a longer RMST yet brain tumor is the shortest, which aligns with the clinical experience in the real-world. By the way, we’d like to kindly note that our paper does not claim that RMST can fully represent invasiveness. Instead, we write “a longer RMST implies weaker invasiveness”.
>
> **Our revisions**: As a response to your question, we have added a detailed description to explain the connect between RMST and tumor invasiveness. It is colored in red and is provided at lines 836-840 in Appendix B.3 on page 16. Please refer to that for the details.
>
> ---
>
> [Reference K] Liu et al., An integrated TCGA pan-cancer clinical data resource to drive high-quality survival outcome analytics. Cell, 2018.
>
> [Reference L] Kim et al., Restricted mean survival time as a measure to interpret clinical trial results. JAMA cardiology, 2017.
>
> ---
>
> Again, we appreciate your invaluable time and constructive suggestions. Thank you so much!

---

### Official Review · Reviewer_SJmc · 2025-11-01

**Soundness:** 3
**Presentation:** 3
**Contribution:** 2
**Rating:** 4
**Confidence:** 5

**Summary:**

This paper conducts the first systematic study on cross-cancer knowledge transfer for WSI-based prognosis prediction, curating a large 26-cancer dataset to demonstrate the feasibility of transferring prognostic knowledge, especially to rare tumors. The authors provide crucial insights into the transfer mechanism by visualizing how transferred models identify novel prognostic regions and statistically analyzing the factors that govern transferability. Finally, the practical utility of this paradigm is validated through a routing-based baseline model, ROUPKT, which effectively leverages multi-cancer knowledge to improve prognostic performance on target tasks.

**Strengths:**

A key strength of the paper is its clear, hypothesis-driven approach. The authors propose that prognostic knowledge can be transferred across different cancer types and that this process is governed by identifiable factors. They then systematically test these hypotheses through a series of well-designed experiments. This structured investigation, which progresses from demonstrating feasibility to exploring underlying mechanisms, provides solid support for the paper's conclusions.

Another significant strength is the clarity of the writing and presentation. The paper is well-structured, logically progressing from problem formulation to experimental results and analysis. The authors effectively use figures and tables to convey complex results in an accessible manner, which makes the paper's novel contributions easy to follow and understand.

**Weaknesses:**

1.  **Limited Generalizability Due to Reliance on a Single Foundation Model:** The study's conclusions are entirely contingent on the feature space of a single foundation model, UNI2-h. The observed transferability patterns (Figure 2) and the predictive power of inter-task factors might be specific artifacts of UNI2-h's architecture and training data. This limits the generalizability of the core findings.
    *   **Actionable Suggestion:** To strengthen the claims, the authors should validate their key findings (e.g., the transferability for a few cancer pairs) using features from at least one other SOTA foundation model (e.g., CTransPath). Minimally, this limitation must be thoroughly discussed.

2.  **Weak Baselines Obscure the True Benefit of Knowledge Transfer:** The main comparison in Table 2 is against simple baselines: a target-specific model ($M_T$) and a fine-tuned version ($E_T$ + MLP). These models do not represent the current state-of-the-art for WSI-based survival analysis, many of which now employ more sophisticated architectures (e.g., graph-based or transformer-based MIL). By comparing `ROUPKT` only to these simple baselines, the paper fails to demonstrate whether cross-cancer knowledge provides a genuine advantage over a more powerful, SOTA target-specific model. The reported 3.1% improvement might diminish or disappear when compared against a stronger baseline.
    *   **Actionable Suggestion:** The authors should implement and compare `ROUPKT` against at least one recent, high-performing SOTA model for WSI survival prediction (e.g., HVT-Surv or a GCN-based model). This would provide a much more convincing measure of the "true" performance gain attributable to knowledge transfer.

3.  **Insufficient Investigation into Negative Transfer:** The paper identifies negative transfer but stops short of providing a meaningful explanation, attributing it simply to an "intrinsic generalization gap." Understanding *why* transfer fails is critical for future work on multi-task learning or task grouping.
    *   **Actionable Suggestion:** The authors should expand their analysis to hypothesize *why* these specific transfers fail, perhaps by analyzing the histopathological uniqueness of tumors like SARC or by visualizing their feature space to show they are significant outliers.

4.  **Missed Opportunity to Evaluate in Few-Shot Scenarios:** The paper's motivation heavily relies on the challenge of modeling rare tumors, which is inherently a low-data problem. However, the experiments are conducted on target datasets with hundreds of samples (e.g., BLCA has N=372). While this demonstrates general applicability, it fails to evaluate the paradigm in the most critical, few-shot learning scenarios where knowledge transfer is expected to be most impactful.
    *   **Actionable Suggestion:** The authors should add experiments that simulate a few-shot setting. This could be done by creating subsets of the existing target datasets with a small number of training samples (e.g., N=20, 50) and evaluating how much knowledge transfer (via `ROUPKT`) improves performance compared to a target-specific model trained on the same limited data. This would directly validate one of the paper's core motivations.

5.  **Methodological Concern in Baseline Selection:** In Table 2, the baselines $M_{S \to T}$ and $E_{S \to T}$ + MLP are based on an "oracle" selection of the best-performing source model on the test set. This inflates the baseline's performance and is not a realistic scenario.
    *   **Actionable Suggestion:** The authors must clarify this oracle setup. A more rigorous evaluation should report a baseline where the source model is selected using a dedicated validation set.

**Questions:**

1.  **On the Generalizability of Findings:** Your study's conclusions are based entirely on features from the UNI2-h model. Could you please clarify how confident you are that the observed transferability patterns (e.g., which cancers are good sources/targets) are fundamental to cancer histology, rather than being specific to the UNI2-h feature space? Would you expect similar results with other foundation models like CTransPath? A response here could address the concern about the generalizability of your core claims.

2.  **On the Strength of Baselines and True Performance Gain:** In Table 2, the performance gain of ROUPKT is measured against relatively simple baselines (ABMIL). How does ROUPKT's performance compare against a more recent, state-of-the-art survival prediction model trained specifically on the target cancer? Clarifying this would help assess the "true" practical benefit of adding cross-cancer knowledge over simply using a more powerful single-task architecture.

3.  **On the Practicality for Rare Tumors (Few-Shot Performance):** A primary motivation for your work is its application to rare tumors with scarce data. However, the experiments were not conducted in a true few-shot setting. Could you provide results or discuss how ROUPKT performs when the target task has a very limited number of training samples (e.g., N < 50)? This would directly test the utility of your approach for its intended key application.

4.  **On the "Oracle" Selection of the Best Source Model:** For the single-source transfer baselines in Table 2, you mention selecting the source with the best performance. Was this selection based on the test set? If so, this is an "oracle" setting. Could you please clarify this and perhaps provide results for a more realistic baseline where the source is selected via a validation set? This would ensure a fairer comparison for ROUPKT.

**Details Of Ethics Concerns:**

N/A.

---

> ### Author Response · Authors · 2025-11-28
> **Response to Reviewer SJmc (1/n)**
>
> Dear Reviewer `SJmc`,
>
> Thank you for your invaluable time and effort for this review. We are encouraged by your recognition of this work, *e.g.*, well-designed experiments, structured investigation, and clear writing and representation.
>
> Also, we are grateful for your careful reading, thorough feedback, high-quality review comments, and constructive and insightful suggestions.
>
> We provide a detailed point-to-point response to your questions and concerns, as follows.

---

> ### Author Response · Authors · 2025-11-28
> **Response to Reviewer SJmc (2/n)**
>
> **W1**: Limited Generalizability Due to Reliance on a Single Foundation Model: The study's conclusions are entirely contingent on the feature space of a single foundation model, UNI2-h. The observed transferability patterns (Figure 2) and the predictive power of inter-task factors might be specific artifacts of UNI2-h's architecture and training data. This limits the generalizability of the core findings. To strengthen the claims, the authors should validate their key findings (e.g., the transferability for a few cancer pairs) using features from at least one other SOTA foundation model (e.g., CTransPath). Minimally, this limitation must be thoroughly discussed.
>
> ---
>
> **AW1**: Thanks for your detailed comment and good suggestions.
>
> We understand your concerns about the reliance on a single foundation model and also agree that other SOTA foundation models could strengthen the claims. We have provided a detailed response and revisions to address concerns, as follows.
>
> We adopt UNI2-h in this study, considering the following points:
> - Concerns about data leakage: Some state-of-the-art foundation models in pathology, *e.g.*, CTransPath (Wang *et al.*, 2022), are pretrained using the WSI data from TCGA. Although they have shown remarkable performances across several tasks, a new concern about data leakage arises because this study’s patient cohorts are also based on TCGA and testing on the same data would lead to an unfair model evaluation.
> - UNI2-h has shown to be a top-tier foundation model: UNI2-h is an improved version of UNI (Chen *et al.*, 2024). It is trained on internal, private WSI data, rather than public data like TCGA. This firstly ensures the fairness in performance benchmarking. Secondly, UNI2-h is a top-tier foundation model, ranking top 2 in the leaderboard released by the Mahmood Lab. We believe that it could have fewer specific artifacts than others, according to the philosophy of “*all models are wrong, but some are useful*” by George Box.
>
> Given the above points, UNI2-h is adopted in this study. We believe that, this foundation model, which is pretrained on large-scale internal, private data and has demonstrated better overall performance than most others, could ensure the generalizability of our core findings to some extent.
>
> Nevertheless, we agree that using another foundation model could strength the claims. Thanks for your good suggestion. Thus, we have followed your suggestion to prepare another state-of-the-art foundation model, CONCH (Lu *et al.*, 2024). Concretely, we have prepared CONCH for five cancer types frequently used in survival analysis benchmarking (Chen *et al.*, 2021; Liu *et al.*, 2025; Cui *et al.*, 2025), *i.e.*, BLCA, BRCA, GBMLGG, LUNG, and UCEC. But unfortunately, due to the large size of WSIs, we are still downloading some missing slides (about 1K in total; 100+ left) as of now. Once finished, we will move forward and post the results on the five cancer types here.
>
> **Our revisions**: Moreover, we also agree that a thorough discussion on this limitation should be provided. We have extended our original discussion and added it to Appendix E Limitations, according to your suggestions. Please refer to the red texts at lines 1200-1206 on page 23.
>
> ---
>
> [Reference A] Wang et al., Transformer-based unsupervised contrastive learning for histopathological image classification. Medical Image Analysis, 2022.
>
> [Reference B] Chen et al., Towards a general-purpose foundation model for computational pathology. Nature Medicine, 2024.
>
> [Reference C] Lu et al., A visual-language foundation model for computational pathology. Nature Medicine, 2024.
>
> [Reference D] Chen et al., Whole slide images are 2d point clouds: Context-aware survival prediction using patch-based graph convolutional networks. MICCAI 2021.
>
> [Reference E] Liu et al., Interpretable Vision-Language Survival Analysis with Ordinal Inductive Bias for Computational Pathology. ICLR 2025.
>
> [Reference F] Cui et al., HiLa: Hierarchical Vision-Language Collaboration for Cancer Survival Prediction. MICCAI 2025.

---

> ### Author Response · Authors · 2025-11-28
> **Response to Reviewer SJmc (3/n)**
>
> **W2**: Weak Baselines Obscure the True Benefit of Knowledge Transfer: The main comparison in Table 2 is against simple baselines: a target-specific model and a fine-tuned version. These models do not represent the current state-of-the-art for WSI-based survival analysis, many of which now employ more sophisticated architectures (e.g., graph-based or transformer-based MIL). By comparing ROUPKT only to these simple baselines, the paper fails to demonstrate whether cross-cancer knowledge provides a genuine advantage over a more powerful, SOTA target-specific model. The reported 3.1% improvement might diminish or disappear when compared against a stronger baseline. The authors should implement and compare ROUPKT against at least one recent, high-performing SOTA model for WSI survival prediction (e.g., HVT-Surv or a GCN-based model). This would provide a much more convincing measure of the "true" performance gain attributable to knowledge transfer.
>
> ---
>
> **AW2**: Thank you for your insightful feedback and valuable suggestion.
>
> To address your concerns, we have implemented two representative baselines in WSI-based survival analysis for comprehensive comparisons, *i.e.*, Patch-GCN (a GCN-based model; Chen *et al.*, 2021) and TransMIL (a Transformer-based model; Shao *et al.*, 2021).
>
> We’d like to kindly note that, since any MIL-based models can be **integrated into** the proposed ROUPKT framework, we apply ROUPKT to these models by freezing their MIL encoders and employing these encoders as the experts in ROUPKT. This experiment is conducted to examine the performance gain attributed to knowledge transfer when varying the architecture of experts.
>
> Comparative results are as follows:
>
> **Table 8**: Results of employing other MIL models as the experts in ROUPKT.
> | Model | BRCA | KIPAN | LUNG | GBMLGG | COADREAD | STES | UCEC | HNSC | SKCM | BLCA | LIHC | CESC | SARC | **Avg.** |
> |:--:|:--:|:--:|:--:|:--:|:--:|:--:|:--:|:--:|:--:|:--:|:--:|:--:|:--:|:--:|
> | - w/ **Patch-GCN** models |
> | $\mathcal{M}_{\mathcal{T}}$ | 0.7017 | 0.8181 | 0.5850 | 0.7853 | 0.6675 | 0.6575 | 0.7369 | 0.6019 | 0.5770 | 0.6164 | 0.7487 | 0.6585 | 0.5736 | **0.6714** |
> | ROUPKT | 0.7389 | 0.8083 | 0.5757 | 0.7797 | 0.7076 | 0.6666 | 0.7441 | 0.6402 | 0.5831 | 0.6520 | 0.7528 | 0.6751 | 0.5808 | **0.6850** |
> | - w/ **TransMIL** models |
> | $\mathcal{M}_{\mathcal{T}}$ | 0.6802 | 0.8121 | 0.5669 | 0.7873 | 0.6651 | 0.6582 | 0.7471 | 0.6315 | 0.5786 | 0.6173 | 0.7385 | 0.6365 | 0.5152 | **0.6642** |
> | ROUPKT | 0.7261 | 0.7921 | 0.5636 | 0.7836 | 0.6993 | 0.6492 | 0.7394 | 0.6389 | 0.5849 | 0.6617 | 0.7521 | 0.6961 | 0.5954 | **0.6832** |
>
> From the above results, we can observe that the proposed ROUPKT framework obtains 2.03% and 2.86% performance gains over vanilla Patch-GCN and TransMIL models, respectively. This further confirms that ROUPKT often effectively utilizes knowledge from other cancers to improve prognosis performance, even when applied to other MIL models. In other words, true benefits of knowledge transfer are observed on three representative models, *i.e.*, ABMIL (in Table 2), Patch-GCN and TransMIL (in Table 8).
>
> **Our revisions**: To address your concerns, we have added a new subsection to show the experiments of employing other MIL models as the experts in ROUPKT. The above results in Table 8 are provided in this new subsection. For more details, please refer to Appendix D.4 Other MIL Models as Experts, the texts colored in red on page 21.
>
> ---
>
> [Reference G] Shao et al., TransMIL: Transformer based Correlated Multiple Instance Learning for Whole Slide Image Classification. NeurIPS 2021.

---

> ### Author Response · Authors · 2025-11-28
> **Response to Reviewer SJmc (4/n)**
>
> **W3**: Insufficient Investigation into Negative Transfer: The paper identifies negative transfer but stops short of providing a meaningful explanation, attributing it simply to an "intrinsic generalization gap." Understanding why transfer fails is critical for future work on multi-task learning or task grouping. The authors should expand their analysis to hypothesize why these specific transfers fail, perhaps by analyzing the histopathological uniqueness of tumors like SARC or by visualizing their feature space to show they are significant outliers.
>
> ---
>
> **AW3**: Thank you for your careful reading and thoughtful comments.
>
> We agree that it is important to provide a meaningful explanation for negative transfer. This could help us to understand why transfer fails. In fact, we did this, *i.e.*, investigating what factors affect transfer performance, in Section 5.2 “What factors affect the transfer of WSI-based prognostic knowledge”. We’d like to describe it in Section 4 (near Negative transfer) to make it clearer.
>
> Specifically, from the results in Table 1 and Figure 4, we found that an under-fitted prognosis model, a difficult cancer prognosis task, and a large gap between source and target in data distribution may result in a failed transfer. For example, when the target is SARC, most models from other cancer fails to transfer to it, because
> - the WSI-based prognosis on SARC is relatively challenging (a C-Index of 0.531 obtained by a dedicated model $\mathcal{M}_{\mathcal{T}}$)
> - and other cancers generally have a larger distance from SARC in terms of overall data distribution (refer to Figure 7(a) $\mathcal{T}$ = SARC).
>
> **Our revisions**: To address your concerns, we have made revisions as follows:
> - We have added an explanation, instead of just saying “intrinsic generalization gap”, to make it more concrete. Please refer to the texts colored in red at lines 213-215 in Section 4 on page 4.
> - We have added a paragraph “Case Study on Negative Transfer” that takes SARC as an example to provide a detailed explanation for why the transfer to SARC fails. Please refer to the red texts at lines 940-944 in Appendix C.2 on page 18.
>
> Thank you again for your good suggestion.

---

> ### Author Response · Authors · 2025-11-28
> **Response to Reviewer SJmc (5/n)**
>
> **W4**: Missed Opportunity to Evaluate in Few-Shot Scenarios: The paper's motivation heavily relies on the challenge of modeling rare tumors, which is inherently a low-data problem. However, the experiments are conducted on target datasets with hundreds of samples (e.g., BLCA has N=372). While this demonstrates general applicability, it fails to evaluate the paradigm in the most critical, few-shot learning scenarios where knowledge transfer is expected to be most impactful. The authors should add experiments that simulate a few-shot setting. This could be done by creating subsets of the existing target datasets with a small number of training samples (e.g., N=20, 50) and evaluating how much knowledge transfer (via ROUPKT) improves performance compared to a target-specific model trained on the same limited data. This would directly validate one of the paper's core motivations.
>
> ---
>
> **AW4**: Thank you for your thorough feedback.
>
> Yes, this study is largely motivated by the issue of limited data in pathology, as you mentioned above. So, we measure the performance of transferred models on rare tumor samples, as shown in Figure 2. However, it seems not sufficient because the proposed ROUPKT is a framework for knowledge transfer but is not examined in a low-data scenario where knowledge transfer is highly anticipated. This is our omission in experimental designs. We really appreciate your insightful comment and are very excited about your good suggestion.
>
> According to your suggestion, we have implemented a few-shot scenario where only a subset of data ($N=$ 32, 64, and 96) can be used in training for $\mathcal{T}$. Given the randomness in subset sampling, we run 5 trials and report the median metric for each experiment. In total, there are 1950 runs on 13 datasets for $\mathcal{M}_{\mathcal{T}}$ and ROUPKT. The result of few-shot performance is summarized below.
>
> **Table 9** Performance of $\mathcal{M}_{\mathcal{T}}$ and ROUPKT in few-shot scenarios.
> | Model | BRCA | KIPAN | LUNG | GBMLGG | COADREAD | STES | UCEC | HNSC | SKCM | BLCA | LIHC | CESC | SARC | **Avg.** |
> |:--:|:--:|:--:|:--:|:--:|:--:|:--:|:--:|:--:|:--:|:--:|:--:|:--:|:--:|:--:|
> | - $N=32$ |
> | $\mathcal{M}_{\mathcal{T}}$ | 0.5062 | 0.5163 | 0.5162 | 0.7477 | 0.6191 | 0.5536 | 0.7002 | 0.4652 | 0.4857 | 0.5716 | 0.6348 | 0.5801 | 0.5332 | **0.5715** |
> | ROUPKT | 0.5263 | 0.5994 | 0.5143 | 0.7364 | 0.6026 | 0.5388 | 0.6673 | 0.5109 | 0.4980 | 0.5395 | 0.6684 | 0.6474 | 0.4882 | **0.5798** |
> | - $N=64$ |
> | $\mathcal{M}_{\mathcal{T}}$ | 0.5275 | 0.5639 | 0.5174 | 0.7523 | 0.6261 | 0.5437 | 0.7073 | 0.4691 | 0.5021 | 0.5614 | 0.6542 | 0.6110 | 0.5307 | **0.5821** |
> | ROUPKT | 0.5307 | 0.5963 | 0.5123 | 0.7588 | 0.6152 | 0.5501 | 0.7151 | 0.5360 | 0.5107 | 0.5411 | 0.6878 | 0.6434 | 0.5156 | **0.5933** |
> | - $N=96$ |
> | $\mathcal{M}_{\mathcal{T}}$ | 0.5547 | 0.6255 | 0.5188 | 0.7558 | 0.6274 | 0.5565 | 0.7139 | 0.4694 | 0.5157 | 0.5705 | 0.6937 | 0.6422 | 0.5343 | **0.5983** |
> | ROUPKT | 0.5778 | 0.7104 | 0.5040 | 0.7641 | 0.6169 | 0.5731 | 0.7331 | 0.5358 | 0.5463 | 0.5605 | 0.7163 | 0.6691 | 0.5493 | **0.6197* |
>
> Our analysis for the above results has two folds.
>
> (1) ROUPKT often performs better than the cancer-specific model trained on the same limited data. It obtains an overall improvement of 1.45%, 1.92%, and 3.58% over $\mathcal{M}_{\mathcal{T}}$ when $N=$ 32, 64 and 96, respectively. This suggests that knowledge transfer via ROUPKT often improves prognosis performance when available training data is limited ($N<100$).
>
> (2) Nevertheless, we also note that on some tasks like LUNG and BLCA, ROUPKT cannot obtain better performance in few-shot learning. One possible reason is that ROUPKT includes a router for expert selecting and a simple adapter for each frozen MIL encoder, which may require more data to train when the target task is relatively difficult. Combining with the approach tailored for few-shot scenarios, *e.g.*, meta-learning (Munkhdalai *et al.*, 2017; Hospedales *et al.*, 2021), could be one possible solution. We’d like to leave it as our future work.
>
> **Our revisions**: As a response to your concerns, we have added a new subsection “Few-Shot Performance” and have presented the experimental results above. For more details, please refer to our revised paper, Table 9 and the red texts in Appendix D.5 Few-Shot Performance on pages 21 and 22.
>
> Thank you again for your insightful comments. We are so lucky to have such a professional reviewer like you. Hats off!
>
> ---
>
> [Reference H] Munkhdalai et al., Meta Networks. ICML 2017.
>
> [Reference I] Hospedales et al., Meta-learning in neural networks: A survey. IEEE TPAMI, 2021.

---

> ### Author Response · Authors · 2025-11-28
> **Response to Reviewer SJmc (6/n)**
>
> **W5**: Methodological Concern in Baseline Selection: In Table 2, the baselines are based on an "oracle" selection of the best-performing source model on the test set. This inflates the baseline's performance and is not a realistic scenario. The authors must clarify this oracle setup. A more rigorous evaluation should report a baseline where the source model is selected using a dedicated validation set.
>
> ---
>
> **AW5**: Thank you for kindly pointing out this.
>
> Our intention is to compare a classical knowledge transfer baseline (*i.e.*, $\mathcal{M}_{\mathcal{S}\to\mathcal{T}}$) with ROUPKT, so we pick the strongest, *i.e.*, a source with the best transfer performance. However, as you pointed out, this is indeed not a realistic scenario.
>
> Therefore, we have reset the criterion for source model selection, according to your suggestion. Namely, we select the source model for each target based on its performance on the validation set of each target. With this new criterion, four targets (BRCA, UCEC, HNSC, and LIHC) have their best source models changed, leading to the degraded overall performances of $\mathcal{M}_{\mathcal{S}\to\mathcal{T}}$ and + MLP.
>
> **Our Revisions**:
> - Table 2: We have updated the results of $\mathcal{M}_{\mathcal{S}\to\mathcal{T}}$ and + MLP  in Table 2, according to the new criterion for source model selection that you mentioned above. Please refer to the colored texts in Table 2 on page 9 for revision details.
> - Textual description about source model selection: We have revised the description regarding source model selection accordingly. Please see the red texts at line 419-420 on page 8.

---

> ### Author Response · Authors · 2025-11-28
> **Response to Reviewer SJmc (7/n)**
>
> **Q1**: On the Generalizability of Findings: Your study's conclusions are based entirely on features from the UNI2-h model. Could you please clarify how confident you are that the observed transferability patterns (e.g., which cancers are good sources/targets) are fundamental to cancer histology, rather than being specific to the UNI2-h feature space? Would you expect similar results with other foundation models like CTransPath? A response here could address the concern about the generalizability of your core claims.
>
> **AQ1**: As clarified in AW1, UNI2-h is an improved version of UNI, as a top-tier foundation model pretrained on large-scale, private WSI data (not TCGA). It has demonstrated remarkable performances on many downstream tasks and ranks top 2 in the leaderboard released by the Mahmood Lab. Given these, we believe that UNI2-h could be close to the “useful” one, given that all models have specifical artifacts or certain limitations.
>
> Other than that, we have prepared CONCH for five cancer types and intend to conduct experiments on them to investigate the transfer patterns on a different foundation model. Once finished slide downloading, we will move forward and post the results here to address your concerns.
>
> Lastly, we’d like to clarify that this study is a preliminary study that investigates for the first time the cross-cancer knowledge transfer in WSI-based cancer prognosis. It is far from perfect. Its limitations span the aspect of foundation models (that you mentioned), study cohorts, and backbone network. We discuss them in Appendix E Limitations and highlight the future works to tackle them. We hope this preliminary study could inspire folks to study WSI-based prognosis via a different approach, *i.e.*, cross-cancer knowledge transfer.
>
> ---
>
> **Q2**: On the Strength of Baselines and True Performance Gain: In Table 2, the performance gain of ROUPKT is measured against relatively simple baselines (ABMIL). How does ROUPKT's performance compare against a more recent, state-of-the-art survival prediction model trained specifically on the target cancer? Clarifying this would help assess the "true" practical benefit of adding cross-cancer knowledge over simply using a more powerful single-task architecture.
>
> **AQ2**: Thanks again for your good suggestion. It is worth noting that ROUPKT is a framework that can integrate with any MIL-based models. It is devised to leverage the knowledge from various cancer sources to benefit a target task.
>
> Therefore, we implement two representative models commonly-used in WSI-based survival analysis, *i.e.*, Patch-GCN and TransMIL, as well as their ROUPKT-based counterparts without changing any hyper-parameters. As shown in AW2, our additional experiments verify the performance improvement of ROUPKT over these models. Given the impressive result obtained by ROUPKT on three different MIL models (ABMIL, Patch-GCN, and TransMIL), we can find that cross-cancer knowledge transfer is a promising approach for enhancing the performance of WSI-based cancer prognosis.
>
> ---
>
> **Q3**: On the Practicality for Rare Tumors (Few-Shot Performance): A primary motivation for your work is its application to rare tumors with scarce data. However, the experiments were not conducted in a true few-shot setting. Could you provide results or discuss how ROUPKT performs when the target task has a very limited number of training samples (e.g., N < 50)? This would directly test the utility of your approach for its intended key application.
>
> **AQ3**: Yes, we have conducted additional few-shot experiments according to your thoughtful feedback. As presented in AW4, we observe that when only few samples ($N=$ 32, 64 and 96) can be used in training, knowledge transfer with ROUPKT still performs better than the cancer-specific model trained on the same limited samples.
>
> We are grateful for your insightful feedback. It does help to strengthen this work.
>
> ---
>
> **Q4**: On the "Oracle" Selection of the Best Source Model: For the single-source transfer baselines in Table 2, you mention selecting the source with the best performance. Was this selection based on the test set? If so, this is an "oracle" setting. Could you please clarify this and perhaps provide results for a more realistic baseline where the source is selected via a validation set? This would ensure a fairer comparison for ROUPKT.
>
> **AQ3**: Yes, it was based the test set. We have realized that it is not a realistic scenario after carefully thinking about your comments. Therefore, we have selected a new source model based on its performance on the validation set of a target. We have updated the results of two transfer baselines using new source models. Moreover, the description about source model selection has been revised accordingly. All revisions are colored in red in our revised paper.
>
> ---
>
> We really feel so fortunate to have such high-quality feedback from you. Thank you so much for your invaluable time and efforts!

---

### Author Response · Authors · 2025-11-22

Dear reviewers,

Here, we would like to express our sincere gratitude for the invaluable time and effort you have dedicated as a reviewer for this work. We are so fortunate to have such high-quality review comments from you, especially in today's special environment. Your constructive suggestions and insightful feedback motivate us to try our best to improve this work further. We are now conducting intensive experiments to address your concerns. Rebuttals will be posted in the next few days, ASAP.

Thanks for your patience.

Best regards

Authors of Submission7252

---

### Author Response · Authors · 2025-11-28

Dear reviewers,

We have posted **our responses** to your concerns and questions and have uploaded **our revised paper**. Thanks for your patience.

As the rebuttal period will be closed within one week, we respectfully ask you to read them. We look forward to receiving feedback from you. If you have further concerns or questions, feel free to raise them. We would be happy to respond promptly.

Thank you again for your invaluable time and high-quality comments!

Best regards

Authors of Submission7252

---

### Author Response · Authors · 2025-12-02
**Messages to AC, SAC, and PCs (3/3)**

*Continue from Above*

**(3) Reviwer `oPrr`**

Reviewer `oPrr` acknowledged this work’s extensive experiments, insightful analysis, novel and interesting method, and excellent contribution.

His/her concerns mainly lie in writing quality and comparative baselines. We summarize the major concerns and our response & revisions below.
- **Writing quality**
  - [Reviewer comment] The writing quality should be improved. The authors are encouraged to work closely with an LLM to address it.
  - [Our response] We have utilized LLM to help us polish our writing and correct grammatical errors. All revisions have been colored in red in our revised paper. Yet, we believe these possible flaws could not usually cause too much difficulty in reading and understanding our paper, as reflected by the other three reviewers’ recognition of this work’s writing and representation.
- **Baseline for positive transfer**
  - [Reviewer comment] Positive transfer should be compared with random initialization and/or mean pooling.
  - [Our response] We have clarified that our criterion, C-Index=0.5, represents sufficiently random prediction, and it is essentially a converged version of random initialization in our scenario.
- **Comparison with a slide foundation model**
  - [Reviewer comment] While a slide foundation model, Feather, was only evaluated in classification tasks, it should also be benchmarked against.
  - [Our response] We have compared our framework with the two baselines based on Feather. Results show that our framework surpasses them by a large margin in overall performance. We have incorporated these results and the corresponding analysis into our revised paper.

In addition, the reviewer also raised some minor concerns & questions, *e.g.*, clarification for C-Index cutoff, and experimental details of Figure 2. Moreover, the reviewer gave many constructive suggestions. We really appreciate his/her careful reading and responsible attitude. We have made a detailed point-to-point response to them and have carefully revised our paper according to each suggestion from the reviewer.

**(4) Reviwer `FJ4V`**

Reviewer `FJ4V` acknowledged this work’s timely problem framing, systematic evaluation, pragmatic method, and excellent presentation.

His/her concerns mainly lie in the appropriateness of the statistical model used for our multivariate analysis, few-shot experiments, and efficiency claims. We summarize the major concerns and our response & revisions below.
- **OLS appropriateness**
  - [Reviewer comment] Using OLS for multivariate analysis likely violates OLS assumptions when C-Index is adopted as a response variable.
  - [Our response] We have changed the statistical model to the Beta regression (tailored for a response variable bounded in [0, 1]). The new results of multivariate analysis still show that four factors help to explain the variations of transfer performance. These results have been added to our revised paper.
- **Few-shot experiments**
  - [Reviewer comment] The authors should compare ROUPKT against simple few-shot fine-tuning or adapters on the target, to contextualize zero-shot vs few-shot practicality.
  - [Our response] We have conducted few-shot experiments. Results show that our framework is still superior to existing baselines when the training data is very limited. This experiment and corresponding analysis have been added to our revised paper.
- **Efficiency claims**
  - [Reviewer comment] Claims of efficiency vs MTL should be supported with actual training time, memory, and FLOPs.
  - [Our response] We have clarified that there is a more fundamental gap between our framework and MTL baselines in hardware requirements. Due to the gigapixel size of WSIs, MTL often has significantly higher requirements than our single-task learning framework. As a response, we have thoroughly discussed and analyzed this issue in the Limitations section.

Thank you again for your invaluable time and effort in the ICLR 2026 review process.

Best regards

Authors of Submission7252

---

### Author Response · Authors · 2025-12-02
**Messages to AC, SAC, and PCs (2/3)**

*Continue from Above*

**(B) A summary of the reviewer’s concerns and our response**

**(1) Reviewer `SJmc`**

Reviewer `SJmc` acknowledged this work’s clear writing & presentation, well-designed experiments, structured investigation, and novel contributions.

His/her concerns mainly lie in experimental designs. We summarize the major concerns and our response & revisions below.
- **Only a single foundation model**
  - [Reviewer comment] Only using a single foundation model (UNI2-h) could limit the generalizability of core findings. The authors should use another model or, at a minimum, should discuss this limitation.
  - [Our response] We clarified the reason why we adopt UNI2-h. As suggested, we have prepared another foundation model for experiments and have thoroughly discussed this limitation in our revised paper.
- **Weak baselines**
  - [Reviewer comment] Some state-of-the-art models with more sophisticated architectures should be compared.
  - [Our response] We have implemented two representative models. Results show that our framework can often further improve these models’ performance. We have incorporated this additional comparison into our revised paper.
- **Few-Shot experiments**
  - [Reviewer comment] Few-shot scenarios should be simulated to support this paper’s core motivation.
  - [Our response] We have conducted few-shot experiments. Results show that our framework is still superior to existing baselines when the training data is very limited. This experiment and corresponding analysis have been added to our revised paper.

Besides the above, the reviewer also raised some minor concerns, *e.g.*, insufficient analysis on negative transfer, and oracle source model selection. We have made a point-to-point response to them and have carefully revised our paper according to the reviewer’s suggestions.

**(2) Reviwer `byot`**

Reviewer `byot` acknowledged this work’s clear writing & presentation, natural solution, and crucial experimental results.

His/her concerns mainly lie in experimental designs (*e.g.*, baselines and metrics) and analysis. We summarize the major concerns and our response & revisions below.
- **Only using ABMIL as the baseline**
  - [Reviewer comment] Comparing with more state-of-the-art baselines could make the result more convincing.
  - [Our response] We have implemented the two most representative models in WSI-based survival analysis. Results show that our framework could often improve these models’ performance further. We have incorporated this additional comparison and corresponding analysis into our revised paper.
- **Criterion for positive/negative transfer**
  - [Reviewer comment] Using a randomly initialized model or random predictions is more rigorous to determine positive/negative transfer.
  - [Our response] We have clarified that our criterion, C-Index=0.5, represents sufficiently random prediction and it is essentially a converged version of the two methods above in our scenario.
- **Comparisons with more MIL networks**
  - [Reviewer comment] The author should add some MIL networks to Table 3 for more convincing comparisons.
  - [Our response] We have clarified that Table 3 is not a classical MIL scenario in which over 1,000 or 10,000 instances are aggregated into a bag-level presentation. The purpose of Table 3 is to compare diverse mechanisms that can utilize 12 frozen MIL encoders.

In addition, the reviewer also raised some minor concerns, *e.g.*, explanations for similar and useful regions, and comparisons with both frozen and fine-tuning baselines. We have made a detailed point-to-point response to them and have carefully revised our paper according to the reviewer’s suggestions.

*Continue Down*

---

### Author Response · Authors · 2025-12-02
**Messages to AC, SAC, and PCs (1/3)**

Dear AC, SAC, and PCs,

We hope you are doing well. We are aware that the incident that happened a few days ago has caused serious impacts. Unfortunately, our ICLR community is going through a difficult time.

As the authors, at first, we’d like to take this opportunity to express our sincere gratitude to PCs for their timely actions and efforts in this incident. Besides, we know that AC will take responsibility for reviewing the paper & response. Thus, we’d also like to thank AC for your subsequent efforts.

Considering the AC’s heavy workload, next, we will
- briefly describe this submission’s situation affected by this incident (part `A`);
- give a summary of the reviewer’s concerns and our response to them (part `B`).

---

**(A) A brief description of this submission’s situation**

To be frank, this paper received very professional, constructive, and insightful comments. We really appreciate the reviewers’ high-quality feedback, so we spent two weeks carefully preparing our response and revising our paper. But unfortunately, the incident happened when we were going to post them.

As a result, we have neither the opportunity to receive further feedback from reviewers nor the chance to discuss with them. That’s so heartbreaking.

---

*Continue Down*

---

### Meta-Review · Area_Chair_i43B · 2026-01-02

**Summary:**

This paper presents a systematic study of cross-cancer knowledge transfer for WSI-based prognosis prediction, supported by a large curated dataset and extensive experiments. Reviewers consistently acknowledged the paper’s clarity, strong empirical effort, and the relevance of the problem, particularly for rare cancers. However, several concerns collectively informed a negative recommendation.

The primary issue is that the central claims remain insufficiently supported at the level expected for ICLR, despite substantial revisions. The study’s conclusions rely heavily on a single foundation model (UNI2-h), leaving open whether the observed transferability patterns and derived insights are model-specific rather than fundamental to histopathology. While the authors discuss this limitation and indicate ongoing experiments with alternative models, the lack of completed cross-model validation significantly limits generalizability.

Additionally, although stronger MIL baselines and few-shot experiments were added in the rebuttal, the empirical gains remain relatively modest and inconsistent across cancers, with clear cases where transfer does not help. As a result, it remains unclear whether the proposed framework provides a decisive advantage over carefully tuned, cancer-specific models, particularly given the added complexity of routing and expert selection.

Finally, some methodological choices—such as the definition of positive/negative transfer and the reliance on C-index thresholds—remain debatable and were defended primarily through argumentation rather than empirical validation. Taken together, these issues prevent the paper from making a sufficiently strong and general claim for acceptance at ICLR.

**Reviewer Concerns:**

The rebuttal is thorough and constructive, and it does address many individual reviewer comments:
* Additional baselines (Patch-GCN, TransMIL) and frozen vs. fine-tuned comparisons were included.
* Few-shot experiments were added and aligned better with the rare-tumor motivation.
* Oracle source selection was corrected to use validation data.
* Statistical analysis was improved by replacing OLS with Beta regression.
* Negative transfer was discussed in more depth with a concrete case study.

However, key concerns remain outstanding:
* The lack of completed experiments with alternative foundation models means the core conclusions are still tied to UNI2-h.
* The added experiments, while helpful, do not fully resolve doubts about the practical significance and robustness of the reported improvements.
* Some conceptual questions (e.g., transfer criteria and failure modes) remain only partially answered.

Overall, while the rebuttal strengthens the paper, it does not fully close the gap to the conference acceptance threshold.

**Reviewer Scores:**

Reviewer SJmc: Likely unchanged or slight increase, but still below acceptance due to remaining generalization concerns.
Reviewer byot: Likely unchanged; core concerns about rigor and transfer criteria persist.
Other reviewers: Likely unchanged, as their original assessments were already cautious or marginal.

---

### Decision · Program_Chairs · 2026-01-26

Reject